



# Drivers of Pine Island Glacier retreat from 1996 to 2016

Jan De Rydt[1], Ronja Reese[2], Fernando Paolo[3], and G. Hilmar Gudmundsson[1]

[1]Department of Geography and Environmental Sciences, Northumbria University, Newcastle upon Tyne, UK
[2]Potsdam Institute for Climate Impact Research (PIK), Member of the Leibniz Association, Potsdam, Germany
[3]Jet Propulsion Laboratory, California Institute of Technology, Pasadena, CA, USA

**Correspondence:** Jan De Rydt (jan.rydt@northumbria.ac.uk)

**Abstract.** Pine Island Glacier in West Antarctica is among the fastest changing glaciers worldwide. Over the last two decades, the glacier has lost in excess of a trillion tons of ice, or the equivalent of 3 mm of sea level rise. The ongoing changes are commonly attributed to ocean-induced thinning of its floating ice shelf and the associated reduction in buttressing forces. However, other drivers of change such as large-scale calving, changes in ice rheology and basal slipperiness could play a vital,
yet unquantified, role in controlling the ongoing and future evolution of the glacier. In addition, recent studies have shown that mechanical properties of the bed are key to explaining the observed speed-up. Here we used a combination of the latest remote sensing datasets between 1996 and 2016, data assimilation tools and numerical perturbation experiments to quantify the relative importance of all processes in driving the recent changes in Pine Island Glacier dynamics. We show that (1) calving and ice shelf thinning have caused a comparable reduction in ice-shelf buttressing over the past two decades, that (2) simulated
changes in ice flow over a viscously deforming bed are only compatible with observations if large and widespread changes in ice viscosity and/or basal slipperiness are taken into account, and that (3) a spatially varying, predominantly plastic bed rheology can closely reproduce observed changes in flow without marked variations in ice-internal and basal properties. Our results demonstrate that in addition to its evolving ice thickness, calving processes and a heterogeneous bed rheology play a key role in the contemporary evolution of Pine Island Glacier.

## 1   Introduction and motivation

Since the 1990s, satellite measurements have comprehensively documented the sustained acceleration in ice discharge across the grounding line of Pine Island Glacier (PIG, Fig. 1) in West Antarctica (Rignot et al., 2002; Rignot, 2008; Rignot et al., 2011; Mouginot et al., 2014; Gardner et al., 2018; Mouginot et al., 2019b). The changes in flow speed are an observable
manifestation of the glacier's dynamic response to both measurable perturbations, such as calving and ice shelf thinning, and poorly constrained variations in physical ice properties and basal sliding. Evidence from indirect observations have indicated that changes in ice shelf thickness have occurred since at least some decades before the 1970s (Jenkins et al., 2010; Smith et al., 2017; Shepherd et al., 2004; Pritchard et al., 2012). Within the last two decades, thinning of the grounded ice (Shepherd





et al., 2001; Pritchard et al., 2009; Bamber and Dawson, 2020), intermittent retreat of the grounding line (Rignot et al., 2014),

changes in calving front position (Arndt et al., 2018) and the partial loss of ice shelf integrity (Alley et al., 2019) have all been reported in considerable detail. At the same time, numerical simulations of ice flow have confirmed the strong link between ice-shelf thinning, which reduces the buttressing forces, and the increased discharge across the grounding line (Schmeltz et al., 2002; Payne et al., 2004; Favier et al., 2014; Arthern and Williams, 2017; Reese et al., 2018; Gudmundsson et al., 2019). Due to the dynamic connection between ocean-driven ice shelf melt rates and tropical climate variability (Steig et al.,

2012; Dutrieux et al., 2014; Jenkins et al., 2016; Paolo et al., 2018), model studies have primarily focused on the important problem of simulating the response of PIG to a potential anthropogenic intensification of melt. Such external perturbations, in combination with ice-internal feedbacks including the Marine Ice Sheet Instability, can force PIG along an unstable and potentially irreversible trajectory of mass loss (Favier et al., 2014; Rosier et al., 2020). Whereas significant progress has been made in simulating the melt-driven retreat of PIG, less attention has been given to other processes that could affect the force

balance and thereby inhibit changes in ice dynamics. Increased damage in the shear margins of the ice shelf, for example, has been reported by Alley et al. (2019) and is known to reduce the buttressing capacity of an ice shelf (Sun et al., 2017). Moreover, a series of recent calving events has caused a sizeable reduction in the extent of the ice shelf, and caused a potential loss of contact with pinning points along the eastern shear margin (Arndt et al., 2018).

The relative impact of changes in ice geometry, basal shear stress and/or ice rheology on the dynamics of PIG has previously

been emphasised in numerical studies by e.g. Schmeltz et al. (2002); Payne et al. (2004); Gillet-Chaulet et al. (2016); Joughin et al. (2019) and Brondex et al. (2019). In all cases, some combination of thickness changes, ice softening, a reduction in ice shelf buttressing or variations in basal shear stress were required to attain an increase in flow speed comparable to observations. Similar conclusions were reached for other Antarctic glaciers. Based on a comprehensive series of model perturbation experiments, Vieli et al. (2007) suggested that the acceleration of the Larsen B Ice Shelf prior to its collapse in 2002 could

not solely be explained by the retreat of the ice shelf front or ice shelf thinning, but required a further significant weakening of the shear margins. Complementary conclusions were reached by Khazendar et al. (2007), who demonstrated the important interdependence of the calving front geometry, a variable ice rheology and flow acceleration based on data assimilation and model experiments for the Larsen B Ice Shelf.

In order to comprehensively diagnose the importance of all processes that have contributed to the acceleration of PIG be-

tween years 1996 and 2016, this study brings together the latest observations and modelling techniques. We consider how calving, ice shelf thinning, the induced dynamic thinning upstream of the grounding line and potential changes in ice-internal and basal properties have caused a different dynamic response across the ice shelf, the glacier's main trunk, the margins and tributaries. Initial observations indicated that the speed-up of PIG was primarily confined to its fast-flowing central trunk (Rignot et al., 2002; Rignot, 2008), though more complex, spatiotemporal patterns of change have emerged more recently (Bamber

and Dawson, 2020). The rapid and spatially diverse acceleration of the flow is an expression of the glacier's dynamic response to changes in the force balance, and it is imperative that numerical ice flow models are capable of reproducing this complex behavior in response to the correct forcing. In general, the driving stress ($\tau_d$), which depends on the ice thickness distribution, is balanced by resistive stresses that include the basal drag ($\tau_b$), side drag through horizontal shear ($\tau_W$), longitudinal resistive



forces ($\tau_\mathrm{L}$) and back forces by the ice shelf ($\tau_\mathrm{IS}$):

$$\tau_\mathrm{d} = \tau_\mathrm{b} + \tau_W + \tau_\mathrm{L} + \tau_\mathrm{IS}\,. \tag{1}$$

It is conceivable that each of the terms in Eq. 1 has changed considerably in recent decades, whilst maintaining a balance at all times. This interplay between different changing forces, in combination with the appropriate boundary conditions, underlie the observed dynamical changes of PIG, and form the backbone of any numerical model simulation. In response to changes in the stress balance, modeled changes in ice velocity between time $t_0$ and $t_1$ can be expanded as follows:

$$\Delta U(\mathbf{x}) \equiv U_{t_1}(\mathbf{x}) - U_{t_0}(\mathbf{x}) = \Delta U_\mathrm{Calv} + \Delta U_\mathrm{Thin} + \Delta U_A + \Delta U_C\,, \tag{2}$$

where terms on the right hand side indicate different contributions to the changes in ice flow, caused by variations in calving front position ($\Delta U_\mathrm{Calv}$), changes in ice thickness ($\Delta U_\mathrm{Thin}$), changes in ice properties commonly parameterized by a rate factor $A$ ($\Delta U_A$), and changes in basal slipperiness $C$ ($\Delta U_C$). Note that these contributions are not generally independent due to feedbacks within the system, and that only the total sum, $\Delta U$, can be observed directly. The velocity component related to

changes in ice thickness, $\Delta U_\mathrm{Thin}$, generally consists of two contributions: an instantaneous response due to ice shelf thinning, as investigated by e.g. Gudmundsson et al. (2019), and the induced dynamic loss and redistribution of mass upstream of the grounding line. We will separately assess the impact of both components. Present-day observations of $\Delta U$ are generally assumed to be dominated by $\Delta U_\mathrm{Thin}$, whereas other contributions remain unquantified and are not generally included in model simulations of future ice flow at decadal to centennial timescales. In particular, temporal changes in ice viscosity and basal

sliding are ignored such that $\Delta U_A + \Delta U_C = 0$, whereas only a minority of ice flow models include (simple) parameterizations of calving. These missing processes, if important, could lead to a systematic bias in model projections of future ice loss, or could prompt the use of unrealistically large perturbations in, e.g., $\tau_\mathrm{IS}$ in an attempt to reproduce observed values of $\Delta U$.

In this study we used a regional configuration of the shallow ice stream flow model, Úa (Gudmundsson, 2020), for PIG to estimate the individual components of $\Delta U$ in Eq. 2 between $t_0 = 1996$ and $t_1 = 2016$. Results enabled us to quantify

the relative importance of each driver of change for the contemporary evolution of PIG, and validate the ability of current-generation ice flow models to reproduce the complex response of PIG to a range of realistic forcings. It is important to note that results were *not* derived from transient simulations of glacier flow based on (uncertain) estimates of the initial model state and external forcings. Instead, the diagnostic model response to a series of *prescribed* changes in ice geometry was analysed, based on the latest observations of ice thinning rates and calving between 1996 and 2016. For each perturbation, changes in

the stress balance (Eq. 1) and the associated ice flow response ($\Delta U_\mathrm{Calv}$ and $\Delta U_\mathrm{Thin}$) were computed. Assuming closure of the velocity budget in Eq. 2 and observed values for $\Delta U$, an estimate for $\Delta U_A + \Delta U_C$ was obtained. Knowledge about the magnitude and spatial distribution of each contribution in Eq. 2 allowed us to verify whether common model assumptions such as $\Delta U_\mathrm{Calv} = 0$ and $\Delta U_A + \Delta U_C = 0$ are indeed justified, at least for contemporary flow conditions.

Although the aforementioned method provides insights into the individual contribution of geometrical perturbations and

changes in ice viscosity and basal slipperiness to overall changes in ice flow, the partitioning between different components of the $\Delta U$ budget likely depends on a number of structural assumptions within the ice flow model. In particular, assumptions



about the form of the basal sliding law are likely to precondition the partitioning of $\Delta U$. Indeed, previous model studies have shown that different forms of the sliding law can produce a distinctly different simulated response of PIG to changes in geometry (Joughin et al., 2010; Gillet-Chaulet et al., 2016; Joughin et al., 2019; Brondex et al., 2019). Based on the assumption

that $\Delta U \approx \Delta U_{\text{Thin}}$, Joughin et al. (2019) showed that a regularized Coulomb law or the plastic limit of a Weertman power-law provide a better fit between modeled and observed changes in surface velocity along the central flowline of PIG, compared to a commonly used viscous Weertman law. Motivated by the above considerations, we explore new ways to derive spatially variable constraints on the form of the sliding law, and thereby provide the first comprehensive, spatially distributed map of basal rheology beneath PIG.

The remainder of this paper is organised as follows. In Sect. 2.1 we introduce the observational datasets used to constrain and validate the ice flow model. Additional details about our data processing methods are provided in App. A. Section 2.2 outlines the experimental design, and provides a summary of the main model components. Further technical details about the model setup and a discussion about the sensitivity of our results to numerical model details are provided in App. B and App. D respectively. Results and an accompanying discussion of all experiments is provided in Sect. 3.1-3.3. Final conclusions are

formulated in Sect. 4.

## 2    Data and methods

The first aim of this study is to simulate the dynamic response of PIG to a series of well-defined geometric perturbations between years 1996 and 2016, and compare model output to observed changes in surface speed over the same time period. As detailed in Sect. 1, geometric perturbations are considered to be observed changes in the calving front position and observed

changes in ice thickness. For each perturbed geometry, a diagnostic solution for the surface velocities, denoted by $U_*$, was obtained, where the subscript $*$ refers to individual perturbations. Since we are primarily interested in the relative contribution of each perturbation to the overall speed-up of PIG between 1996 and 2016, our focus will be on relative changes $\Delta U_*/\Delta U$, where $\Delta U_* = U_* - U_{96}$ and $\Delta U = U_{16} - U_{96}$. In order to compute the relative changes in surface flow, two types of model experiment are required: (1) inverse simulations, which were used to obtain model configurations that are as close as possible

to the observed state of PIG in 1996 ($U_{96}$) and 2016 ($U_{16}$), and (2) perturbation experiments to obtain estimates of $U_*$, starting from the 1996 model configuration. In Sect. 2.1 we list the data sources required for these experiments, whereas a detailed overview of the experimental design is provided in Sect. 2.2.

### 2.1    Observed changes of Pine Island Glacier between 1996 and 2016

Our study area and model domain encompasses the 135,000 km$^2$ grounded catchment (Rignot et al., 2011) and seaward floating

extension of PIG in West Antarctica, as depicted in Fig. 1a. To investigate the physical processes that forced the contemporary speed-up of the glacier, and its increase in grounding line flux between years 1996 and 2016, we required detailed observations of the surface velocity, ice thickness and calving front position for both years.

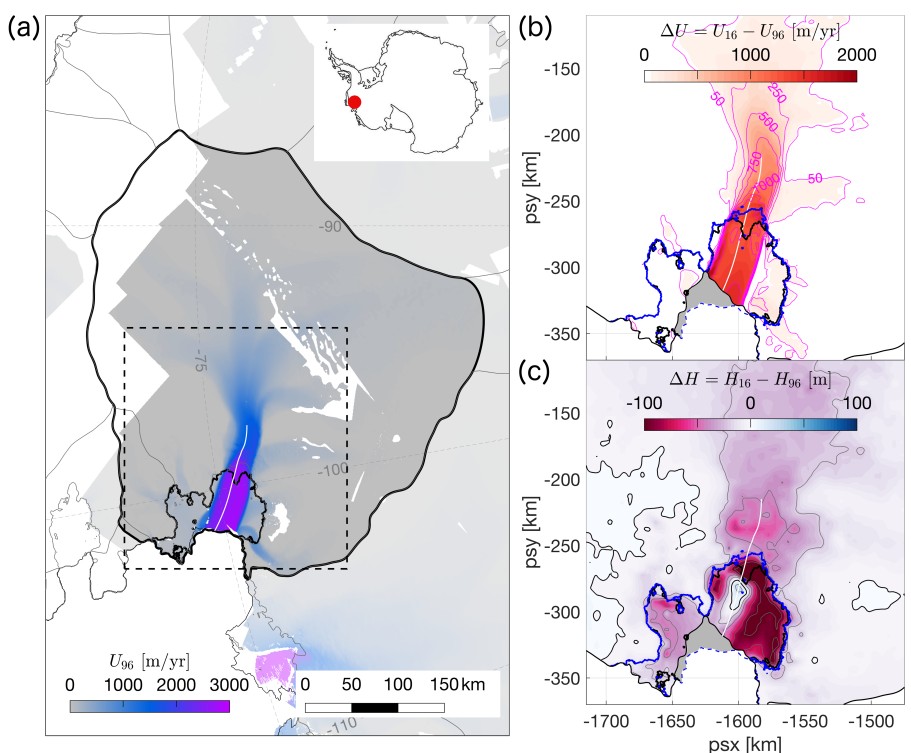

**Figure 1.** Pine Island Glacier (PIG) and its location in West Antarctica. **(a)** Surface speed of PIG in 1996 in m/yr, as reported by the MEaSUREs program (Mouginot et al., 2019a). Solid black outlines delineate the extent of the PIG catchment (Rignot et al., 2011) and 1996 grounding line position (Rignot et al., 2014). The white line along the central flowline indicates the location of the transect in Fig. 2. The dashed rectangle corresponds to the extent of panels **b** and **c**. **(b)** Observed increase in surface speed (Mouginot et al. (2019a), colours and contours in m/yr) and loss of ice shelf extent (grey shaded area) between 1996 and 2016. The blue line indicates the 2011 grounding line (Rignot et al., 2014). **(c)** Total change in ice thickness between 1996 and 2016 ($\Delta H$ in m), based on a combination of CPOM data (Shepherd et al., 2016) for the grounded ice and newly analyzed data for the ice shelf (Appendix A). The zero contour is shown in black, other contours in grey are spaced at 20 m intervals.

The surface velocity measurements used in this study were taken from the MEaSUREs database (Mouginot et al., 2019a, b). For 1996, Synthetic Aperture Radar data from the ERS-1/2 mission were processed using interferometry techniques and

combined into a mosaic with effective timestamp 01/01/1996. The MEaSUREs velocities for 2016 were based on feature tracking of Landsat 8 imagery with effective timestamp 01/01/2016. The change in surface speed between both years (denoted by $\Delta U = U_{16} - U_{96}$) is shown in Fig. 1b, and we refer to e.g. Rignot et al. (2014) and Gardner et al. (2018) for a more comprehensive description of these observations.

Recent estimates of ice thickness were obtained from the BedMachine Antarctica dataset (Morlighem et al., 2020), which

provides both high-resolution surface topography based on the REMA mosaic (Howat et al., 2019) and improved estimates of bedrock topography using mass conservation methods. The nominal date for this dataset corresponds to the date stamp of the





REMA elevation model, which is spatially variable but largely between 2014 and 2018 for PIG. For consistency with previous notation we refer to the BedMachine Antarctica ice thickness as $H_{16}$ and we assume a uniform timestamp of 01/01/2016.

Ice thickness estimates for 1996, henceforth denoted by $H_{96}$, were obtained by subtracting measurements of ice thickness change between 1996 and 2016, denoted by $\Delta H$, from $H_{16}$, i.e. $H_{96} = H_{16} - \Delta H$. Estimates of $\Delta H$ were based on a combination of existing CPOM measurements of thickness change rates (Shepherd et al., 2016) for areas upstream of the 2016 grounding line, and newly analyzed data for the floating ice shelf. Detailed information about the latter can be found in App. A. The resulting values for $\Delta H$, linearly interpolated across the grounding line and in data sparse areas, are shown in Fig. 1c, and provide the most comprehensive observation-based ice shelf and grounded ice thickness changes for PIG to date.

The grounding line location for $H_{16}$ (blue line in Fig. 1b-c) corresponds to the DInSAR derived grounding line in 2011 from Rignot et al. (2014), since this is included as a constraint in the generation of the BedMachine Antarctica bed topography. In addition, localized adjustments less than 150 m were made to the bed topography to ensure that the grounding line for $H_{96}$ (black lines in Fig. 1a-c) corresponds to the DInSAR derived grounding line in 1992-1996 (Rignot et al., 2014).

Alongside the above-listed changes in flow dynamics and ice thickness, the calving front of PIG retreated by up to 30 km between 1996 and 2016 during a succession of large-scale calving events; see e.g. Arndt et al. (2018). We traced the calving front positions in 1996 and 2016 from cloud free Landsat 5 and Landsat 8 panchromatic band images with timestamps 18/02/1997 and 25/12/2016 respectively. Both outlines are included in Fig. 1b-c, and the ice shelf area that was lost between 1996 and 2016 is shaded in grey.

## 2.2 Experimental design

We discuss the numerical experiments required to obtain an optimal model configuration for the state of PIG in 1996 ($U_{96}$) and 2016 ($U_{16}$) in Sect. 2.2.1. Experiments that provide estimates of $U_*$ for a series of observed perturbations in the geometry of PIG are introduced in Sect. 2.2.2. Experiments that simulate changes in the rate factor or basal slipperiness are detailed in Sect. 2.2.3.

### 2.2.1 Inverse experiments

We explicitly solved the stress balance in year 1996 (an analogous routine was applied for 2016) by assimilating the known ice thickness ($H_{96}$), calving front position and surface velocity ($U_{96}$) in the shallow ice stream (SSA) model Úa (Gudmundsson et al., 2012; Gudmundsson, 2020). This 'data assimilation' or 'inverse' step is commonly adopted in glaciology (see MacAyeal (1992) for one of the earliest examples) to minimize the misfit between modeled and observed surface velocities through the optimization of uncertain physical parameters. The inverse capabilities of Úa (further details in App. B) were used to optimize the uncertain spatial distribution of the rate factor, $A$, and basal slipperiness, $C$. These physical parameters define the constitutive model and the relationship between basal shear stress $\tau_b$ and basal sliding velocity $U_b$ respectively:

$$\dot{\epsilon} = A\tau_{\mathrm{E}}^{n-1}\tau, \tag{3}$$

$$\tau_b = C^{-1/m}\|U_b\|^{\frac{1}{m}-1}U_b \tag{4}$$





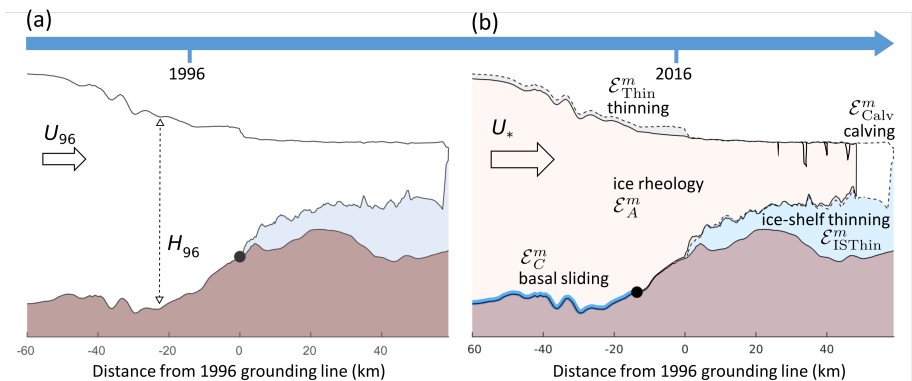

**Figure 2.** Overview of changes along the Pine Island Glacier centerline from (**a**) year 1996 to (**b**) year 2016. Increased ice flow is driven by a combination of calving, ice-shelf thinning and dynamic thinning with movement of the grounding line, as well as changes in basal sliding and ice rheology. Transects of the geometry are based on observations along the flowline indicated in Fig. 1; black dots indicate the respective grounding line positions in both years. Crevasses are introduced for illustration purposes only and do not strictly correspond to observed features. The importance of each 'driver of change' was investigated in a series of numerical perturbation experiments, denoted by $\mathcal{E}_*^m$ in panel **b**, with $m$ indicating the sliding exponent and $*$ the respective experiment described in section 2.2.

Glen's law, Eq. 3, relates the strain rates $\dot{\epsilon}$ to the deviatoric stress tensor $\tau$. A creep exponent $n = 3$ was used throughout this
study. Equation 4 is known as a Weertman sliding law (Weertman, 1957), and describes a linear, viscous or close-to plastic bed rheology for $m = 1$, $m > 1$ and $m \gg 1$ respectively. Throughout this study, a range of values for $m$ are considered, as specified below. For each $m$ we performed a new inversion for $A$ and $C$, which caused small variations in $\tau_b$ between cases, but produced an optimal fit between modeled and observed surface velocities in each case. This method differs from other studies, e.g. Joughin et al. (2019), who performed a single inversion for $m = 1$, and obtained $C$ for different values of $m$ by
solving Eq. 4 under the assumption that $\tau_b$ remains constant. We consider our approach to be more appropriate for this study, as our focus is primarily on an accurate model representation of the surface flow (e.g. Eq. 2). Results for $A$ and $C$ for $m = 3$ are provided in Appendix B. The outcome of the inverse step is a best estimate for each term in Eq. 1, based on observations of geometry and surface velocity of PIG in year 1996. Analogous results were obtained for 2016.

### 2.2.2   Geometric perturbation experiments

In the second step we carried out a series of numerical perturbation experiments, starting from the 1996 model configuration, to simulate the impact of observed changes in geometry on the flow of PIG. The rate factor and basal slipperiness were kept fixed to their 1996 values. For each perturbation, the modified force balance (Eq. 1) and corresponding surface velocities, $U_*$, were diagnosed within Úa. Experiments are referred to as $\mathcal{E}_*^m$ with a variable subscript to indicate the type of perturbation and a superscript to specify the value of the sliding exponent $m$. Experiments were carried out for a range of exponents so we leave
$m$ unspecified for now.





  – $\mathcal{E}_{\mathrm{Calv}}^{m}$. Changes in the calving front location were prescribed to reflect the loss of ice shelf between 1996 and 2016 (see Fig. 1b-c). All model grid elements downstream of the 2016 calving front (grey shaded area in Fig. 1b) were deactivated, whilst elements upstream of the 2016 calving front remained fixed to avoid numerical interpolation errors. All other model variables were kept fixed.

185  – $\mathcal{E}_{\mathrm{ISThin}}^{m}$. Changes in ice shelf thickness were prescribed, corresponding to observed thinning of the ice shelf between 1996 and 2016 (Fig. 1c). Note that the calving front and grounding line location did not change in this experiment. This experiment is similar to previous studies, e.g. (Reese et al., 2018; Gudmundsson et al., 2019).

  – $\mathcal{E}_{\mathrm{Thin}}^{m}$. Observed changes in both the floating and grounded parts of PIG were prescribed. This caused the grounding line to move from its 1996 position (black line in Fig. 1b-c) to the 2016 position (blue line in Fig. 1b-c).

190  – $\mathcal{E}_{\mathrm{CalvThin}}^{m}$. Combined changes in calving front position (as in $\mathcal{E}_{\mathrm{Calv}}^{m}$), and thinning (as in $\mathcal{E}_{\mathrm{Thin}}^{m}$) were prescribed.

A schematic overview of the experiments is provided in Fig. 2. While $\mathcal{E}_{\mathrm{Calv}}^{m}$ allows us to assess the time-integrated impact of calving between 1996 and 2016 ($\Delta U_{\mathrm{Calv}}$), and $\mathcal{E}_{\mathrm{ISThin}}^{m}$ simulates the instantaneous response to total changes in ice thickness between 1996 and 2016 ($\Delta U_{\mathrm{ISThin}}$), both experiments ignore the time-dependent, dynamic response of the upstream grounded ice. These separate perturbations make it possible to disentangle the changes in ice shelf buttressing caused by each process,

and hence their relative importance for driving the transient evolution of the flow. Dynamic thinning of grounded ice, as well as migration of the grounding line, is included in the experiments $\mathcal{E}_{\mathrm{Thin}}^{m}$, which allows us to determine the full response to changes in ice thickness ($\Delta U_{\mathrm{Thin}}$). Finally, the experiment $\mathcal{E}_{\mathrm{CalvThin}}^{m}$ accounts for all geometric perturbation, and provides a spatial distribution of $\Delta U_{\mathrm{Thin}} + U_{\mathrm{Calv}}$.

### 2.2.3 Estimates of changes in $A$ and $C$

Later on we show that geometric perturbations alone are not able to fully reproduce the observed patterns of speed-up across the PIG catchment, i.e. $\Delta U \neq \Delta U_{\mathrm{CalvThin}}$. It is conceivable that, along with the evolving geometry, variations in ice and basal properties have contributed to the changes in flow between 1996 and 2016, i.e. $\Delta U_{A} + \Delta U_{C} \neq 0$. Indeed, feedback mechanisms are likely to cause an important interdependence between geometry-induced changes in ice flow, shear softening and/or changes in basal shear stress. Reliable observations of changes in rheology and basal properties are not available, but

numerical inverse simulations can provide valuable insights into their evolution. We used the inverse method as described in Sect. 2.2.1 and App. B to estimate necessary bounds on the magnitude and spatial distribution of changes in $A$ and $C$ that are required beside the geometrical changes already applied, to produce the speed-up of PIG between 1996 and 2016. Changes in $A$ and $C$ are treated separately.

  – $\mathcal{E}_{A}^{m}$. The aim of this experiment is to determine possible changes in the rate factor between 1996 ($A_{96}$) and 2016 ($A_{16}$).

210  $A_{96}$ was previously obtained in part 1 (inverse step) of the experimental design. To estimate $A_{16}$, an inverse optimization problem was solved for the 2016 PIG geometry ($H_{16}$) and velocities ($U_{16}$), but using a cost function that was minimized with respect to $A$ only. The slipperiness $C$ was kept fixed to its 1996 solution.





- $\mathcal{E}_C^m$. This experiment is analogous to $\mathcal{E}_A^m$, but the cost function in the inverse problem was optimized with respect to $C$ only, whereas the rate factor $A$ was kept fixed to its 1996 solution.

## 3 Results and discussion

### 3.1 Ice dynamics response to changes in geometry between 1996 and 2016

We present results for the first set of perturbation experiments, which simulate the impact of observed changes in geometry on the flow of PIG. As detailed in section 2.2.2, perturbations are split between four separate cases: calving ($\mathcal{E}_{\mathrm{Calv}}^3$), thinning of the ice shelf ($\mathcal{E}_{\mathrm{ISThin}}^3$), thinning of the ice shelf and grounded ice ($\mathcal{E}_{\mathrm{Thin}}^3$), and the combined impact of all observed geometrical changes ($\mathcal{E}_{\mathrm{CalvThin}}^3$). We did not previously specify the value of the sliding exponent, however, here we set $m = 3$, which is a commonly adopted value in ice flow modeling and describes a viscous, rate-strengthening bed rheology. We will explore results for different values of $m$ in Sect. 3.3. Results for the relative change in surface speed for each of the above perturbations are presented in Fig. 3a-d.

Calving as simulated in $\mathcal{E}_{\mathrm{Calv}}^3$ causes changes in flow speed that are predominantly restricted to the ice shelf, where it accounts for up to 50% of the observed speed-up between 1996 and 2016 (Fig. 3a). A smaller dynamical impact is also felt upstream of the grounding line, caused by the calving-induced reduction in ice shelf buttressing and mechanical coupling between the floating and grounded ice. Along the central, fast-flowing trunk of PIG, calving typically accounts for less than 10% of the observed speed-up, with little or no effect on the dynamics of the upstream tributaries. The only area with negative relative changes is the western shear margin of the ice shelf, where modeled and observed changes in flow speed have the opposite sign. Extensive damage has caused this margin to migrate and significant interannual variations in flow speed have been reported by Christianson et al. (2016), a process that is not captured by this experiment.

Flux gates provide an alternative, aggregated way to convey the above results. We present flux calculations for two gates perpendicular to the flow within the central part of PIG, as displayed in Fig. 3a. Gate 1 is situated about 50 km upstream of the 2016 grounding line and captures the inland propagation of changes in ice flow. Gate 2 approximately coincides with the 2016 grounding line position and captures changes in grounding line flux, which is a direct measure for PIG's increasing contribution to sea level rise, and an important indicator of change. Figure 3e shows that calving accounts for 2% and 13% of the observed flux changes through Gate 1 and 2 respectively. This supports the earlier conclusion that the retreat of the PIG calving front between 1996 and 2016 has caused only minor instantaneous changes to the flow upstream of the grounding line.

Thinning of the ice shelf as simulated in experiment $\mathcal{E}_{\mathrm{Thin}}^3$ induces a flow response that is similar to calving, as shown in Fig. 3b, and indicates that calving and ice shelf thinning have caused a similar perturbation in the buttressing forces. The largest percentage changes are found on the ice shelf, and are typically less than 25%, while the relative flux changes through Gate 1 and 2 are identical to the calving experiment (Fig. 3e). Ice shelf thinning is generally accepted to be the main driver of ongoing mass loss of PIG, and patterns of ice shelf thinning elsewhere in Antarctica are strongly correlated to observed changes in grounding line flux (Reese et al., 2018; Gudmundsson et al., 2019). However, the force perturbations that result from ice shelf thinning alone, in particular the instantaneous reduction in back forces $\tau_{\mathrm{IS}}$, are not sufficient to explain the magnitude of





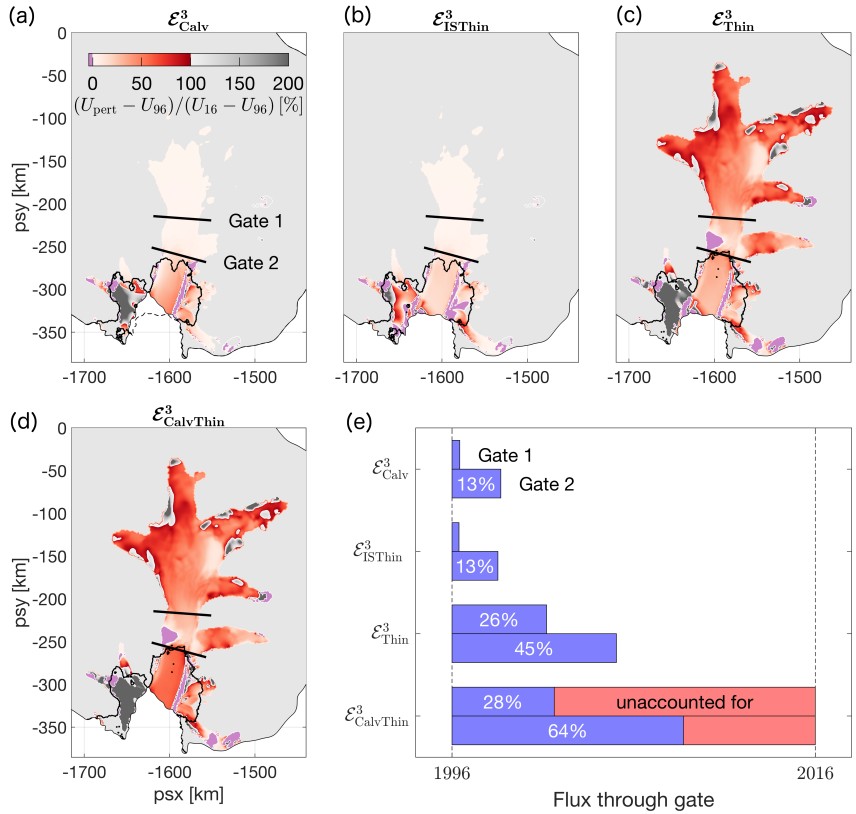

**Figure 3.** Modelled changes in surface speed compared to 1996 for prescribed perturbations of the Pine Island Glacier geometry. **(a)** Retreat of the calving front, **(b)** thinning of the ice shelf, **(c)** thinning of the ice shelf and grounded ice, including grounding line retreat, **(d)** calving and thinning combined. For each perturbation, the modeled change in speed ($U_* - U_{96}$) is expressed as a percentage of the observed speed-up between 1996 and 2016 ($U_{16} - U_{96}$). Panel **(e)** shows the percentage of the observed flux changes through Gate 1 and 2 that can be explained by the respective perturbations. The simulated impact of calving and thinning in experiment $\mathcal{E}^3_{\mathrm{CalvThin}}$ underestimates measured flux changes by 72% and 36% respectively. Possible explanations for the unexplained increase in flow speed are provided in Sect. 3.2 and Sect. 3.3 .

observed changes in upstream flow. Indeed, experiment $\mathcal{E}^3_{\mathrm{ISThin}}$ demonstrates that the direct and instantaneous contribution of ice shelf thinning to observed changes in grounding line flux are less than 25%. Instead, time-evolving changes in geometry and mass redistribution upstream of the grounding line play a significant role in increasing the dynamic response of the glacier. These dynamic changes, caused indirectly by changes in the calving front position and ice shelf thinning, were not captured

by the experiments $\mathcal{E}^3_{\mathrm{Calv}}$ and $\mathcal{E}^3_{\mathrm{ISThin}}$, but are considered in experiment $\mathcal{E}^3_{\mathrm{Thin}}$.

In experiment $\mathcal{E}^3_{\mathrm{Thin}}$ we prescribed the time integrated change in ice thickness between 1996 and 2016 for both the floating ice shelf and upstream grounded ice. This perturbation incorporates the observed recession of the PIG grounding line between 1996 and 2016. The combined reduction in ice shelf buttressing, loss of basal friction due to grounding line retreat and changes in driving stress caused a significant and far-reaching impact on the flow, as displayed in Fig. 3c. Modeled changes on the





ice shelf are consistent with and similar in amplitude to $\mathcal{E}^3_{\text{ISThin}}$. Upstream of the grounding line, modeled changes relative to observations are between 25% and 50% along the central trunk and up to 100% along the tributaries. In addition, results demonstrate that glacier-wide changes in ice thickness account for 26% and 45% of the observed changes in ice flux through Gate 1 and 2 respectively (Fig. 3e).

In the final perturbation experiment, $\mathcal{E}^3_{\text{CalvThin}}$, the combined effect of calving and changes in ice thickness were simulated.
Modeled versus observed changes in surface speed are shown in Fig. 3d. The spatial pattern is consistent with previous experiments, and the amplitude of the response is approximately equal to the added response of experiments $\mathcal{E}^3_{\text{Calv}}$ and $\mathcal{E}^3_{\text{Thin}}$, i.e. $\Delta U_{\text{CalvThin}} \approx \Delta U_{\text{Calv}} + \Delta U_{\text{Thin}}$. The corresponding percentage changes in ice flux through Gate 1 and 2 are 28% and 64% respectively, whereas modeled changes in flow across the grounding line proper account for about 75% of the observed increase in flux between years 1996 and 2016. Although this experiment prescribes all observed changes in PIG geometry over
the observational period, model simulations are unable to capture a significant percentage of the observed speed-up. This is most noticeable along the fast-flowing central trunk upstream of the grounding line, whereas discrepancies decrease along the slow-flowing tributaries in the high catchment. We also note that in one area between Gate 1 and 2, modeled and observed changes in surface speed have opposite signs.

Although it is not unexpected to find differences between diagnostic model output and observations, the consistently sup-
pressed response of the model to realistic perturbations in ice geometry is indicative of a structural shortcoming within our experimental design. Indeed, results show that for a viscous bed rheology described by a Weertman sliding law with constant sliding coefficient $m = 3$, changes in ice geometry alone cannot explain the complex and spatially variable pattern of speed-up over the observational period, i.e. $\Delta U \neq \Delta U_{\text{CalvThin}}$. In the remainder of this study, two possible hypotheses are analyzed that enable to close the gap between geometry-induced changes in ice flow and the observed speed-up of PIG. The
first hypotheses, which is considered in section 3.2, assumes that bed deformation can indeed be described by a viscous power law with $m = 3$, but further temporal variations in ice viscosity and/or basal slipperiness are required in addition to changes in geometry: $\Delta U = \Delta U_{\text{CalvThin}} + \Delta U_A + \Delta U_C$. The second, alternative hypotheses, discussed in section 3.3, assumes that internal properties of the ice and bed have not significantly changed between years 1996 and 2016, i.e. $\Delta U_A + \Delta U_C \approx 0$, but a different physical description of the basal rheology is required instead.

## 3.2 Changes in the rate factor and basal slipperiness between years 1996 and 2016

In transient model simulations of large ice masses such as Antarctica's glaciers and ice streams, it is common to assume that the advection of $A$ with the ice, or changes due to temperature variations and fracture as well as changes in basal slipperiness $C$, exert a second-order control on changes in ice flow. As such, temporal variability in $A$ and $C$ are often ignored, based on the assumption that these changes are sufficiently slow and do not significantly affect the flow on typical decadal to centennial
timescales under consideration. The aim of experiments $\mathcal{E}^3_A$ and $\mathcal{E}^3_C$, as outlined in section 2.2.3, is to establish whether this is a valid assumption, or whether previously ignored changes in $A$ and/or $C$ can provide a realistic explanation for the discrepancy between simulated and observed changes in the surface speed of PIG in the geometric experiment $\mathcal{E}^3_{\text{CalvThin}}$. Experiment $\mathcal{E}^3_A$ assumes that, in addition to changes in geometry, temporal variations in $A$ alone are able to explain the significant increases

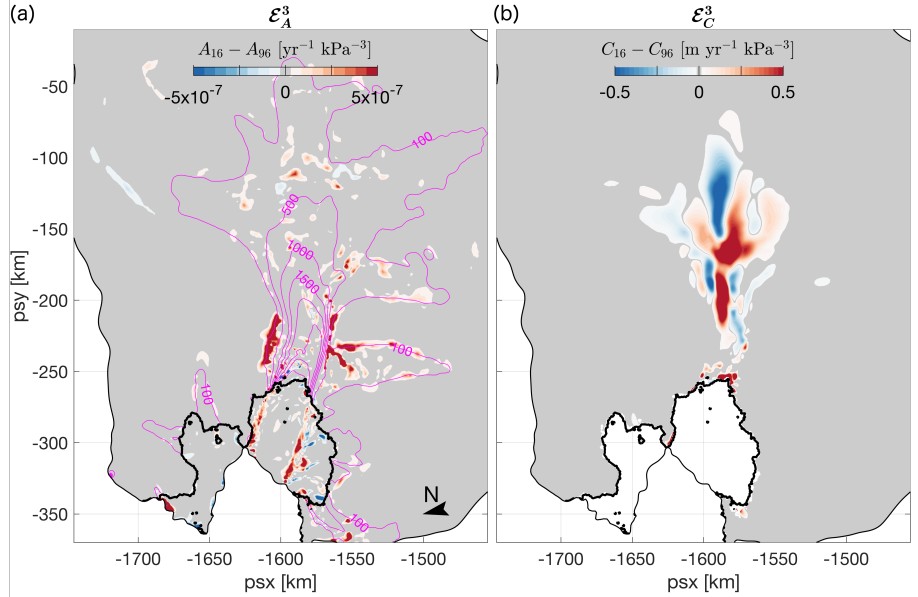

**Figure 4.** (a) Results for the $\mathcal{E}_A^3$ experiment: changes in the rate factor $A$ required to fully explain the observed changes in surface speed of the ice shelf and grounded ice between years 1996 and 2016. The sliding exponent $m = 3$ and basal slipperiness $C$ are kept fixed between 1996 and 2016. Magenta contours (in m/yr) correspond to the surface speed in 2016. (b) Results for the $\mathcal{E}_C^3$ experiment: changes in the basal slipperiness $C$ required to explain the observed increase in surface speed of the grounded ice between 1996 and 2016. The rate factor $A$ is assumed constant between 1996 and 2016.

in flux that were unaccounted for in previous experiments. Alternatively, $\mathcal{E}_C^3$ assumes that, in addition to changes in geometry,

temporal variations in $C$ alone are able to explain the discrepancy in section 3.1 between the modeled and observed speed-up. In line with previous experiments we assume a Weertman sliding law with $m = 3$. The results for both experiments are summarized in Fig. 4.

    Changes in $A$ (Fig. 4a), needed to fully reproduce the speed-up of PIG between years 1996 and 2016, are spatially coherent and predominantly positive. This suggests a reduction in ice viscosity between 1996 and 2016, either as a result of

localized heating, enhanced damage within the ice column or changes in anisotropy. The largest changes are found in distinct geographical areas: a localized increase within the shear margins of the ice shelf, and a more widespread increase along the slower-moving flanks (magenta contours in Fig. 4a indicate surface speed in 2016) of the main glacier and westernmost tributary, about $20\,\mathrm{km}$ upstream of the 2016 grounding line. Changes within the ice shelf shear margins are consistent with their increasingly complex and damaged morphology, as is apparent from satellite images (Alley et al., 2019). Weakening of the ice

in these areas accounts for the remaining 50% of observed changes in ice-shelf speed-up that could not previously be explained by calving and ice shelf thinning alone (experiment $\mathcal{E}_{\mathrm{CalThin}}^3$). Projected changes in $A$ along the flanks of the upstream glacier, on the other hand, are more ambiguous. Values in excess of $10^{-7}\,\mathrm{yr}^{-1}\mathrm{kPa}^{-3}$ correspond to an equivalent increase in 'ice' temperature by up to $40\,^\circ\mathrm{C}$. This is nonphysical unless (part of) the change is attributed to damage or evolution of the ice





fabric. Based on our analysis of Sentinel and Landsat satellite images, there is no obvious indication of recent changes in the
surface morphology in these areas. Either significant and wide-spread changes in the thermal and mechanical properties have
occurred beneath the surface, or the observed speed-up and thinning in these areas, as previously reported by Bamber and
Dawson (2020), cannot be convincingly attributed to changes in the rate factor.

Alternatively, temporal changes in $C$ can be invoked to explain the discrepancies between modeled and observed changes
in surface speed between years 1996 and 2016. Results presented in Fig. 4b suggest that a complex and widespread pattern
of changes in the slipperiness is required across an extensive portion of PIG's central basin and its upper catchment. Despite
the complex and poorly understood relationship between $C$ and quantifiable physical properties of the ice/bed interface, it is
difficult to understand how any single process or combination of physical processes could be responsible for the large and
widespread changes in $C$ over a time period of two decades. Further information, such as a timeseries of maps similar to
Fig. 4b, can potentially be used to test the robustness of this result and provide further insights into the physical processes that
could control such changes. This is the subject of future research.

We note that in the $\mathcal{E}_C^3$ experiment, velocities on the floating ice shelf were largely unaffected by changes in $C$, and remained
significantly slower than observations (not shown). In contrast, changes in the rate factor were able to fully account for the
speed-up of the ice shelf. On the other hand, large variations in $A$ were needed to explain the changes in ice dynamics along the
slow-moving flanks of PIG (Fig. 4a), whereas only small changes in $C$ less than $10^{-3} \, \mathrm{yr}^{-1} \mathrm{kPa}^{-3} \mathrm{m}$ were required to explain
this behaviour. It is therefore conceivable that, in addition to PIG's evolving geometry, an intricate combination of changes
in both the rate factor and basal slipperiness are required to explain the glacier's complex and spatially-diverse patterns of
speed-up over the last two decades. It is however not straightforward to disentangle these processes in the current modeling
framework.

### 3.3 Evidence for a heterogeneous bed rheology

The relationship between changes in geometry and the dynamic response of a glacier crucially depends on the mechanical
properties of the underlying bed and subglacial hydrology. So far, we have assumed that basal sliding can be represented by
a viscous power law with spatially uniform stress exponent $m = 3$ (see Eq. 4). A viscous rheology is particularly suitable
for the description of hard-bedded sliding without cavitation, but missing processes such as variations in effective pressure or
the deformation of a subglacial till layer with a maximum shear (yield) stress could be important limitations. Some evidence
has been provided for plastic bed properties underneath ice streams either from observations (Tulaczyk et al., 2000; Minchew
et al., 2016) or laboratory experiments (Zoet and Iverson, 2020). Most recently, Gillet-Chaulet et al. (2016), Brondex et al.
(2019) and Joughin et al. (2019) used numerical simulations to show that different sliding laws can cause a distinctly different
dynamical response of PIG to changes in geometry, and observed changes in surface velocity were best reproduced for sliding
exponents $m \gg 1$ or using a hybrid law that combines Weertman with Coulomb sliding. Although the results are compatible
with a plastic bed underlying the central trunk of PIG, no constraints on the spatial variability in basal rheology were derived.

In order to quantify how different values of the sliding exponent affect the sensitivity of PIG to changes in geometry across
the catchment, we repeated perturbation experiments $\mathcal{E}_{\mathrm{CalvThin}}^m$ for a range of sliding law exponents, from $m = 1$ to $m = 21$

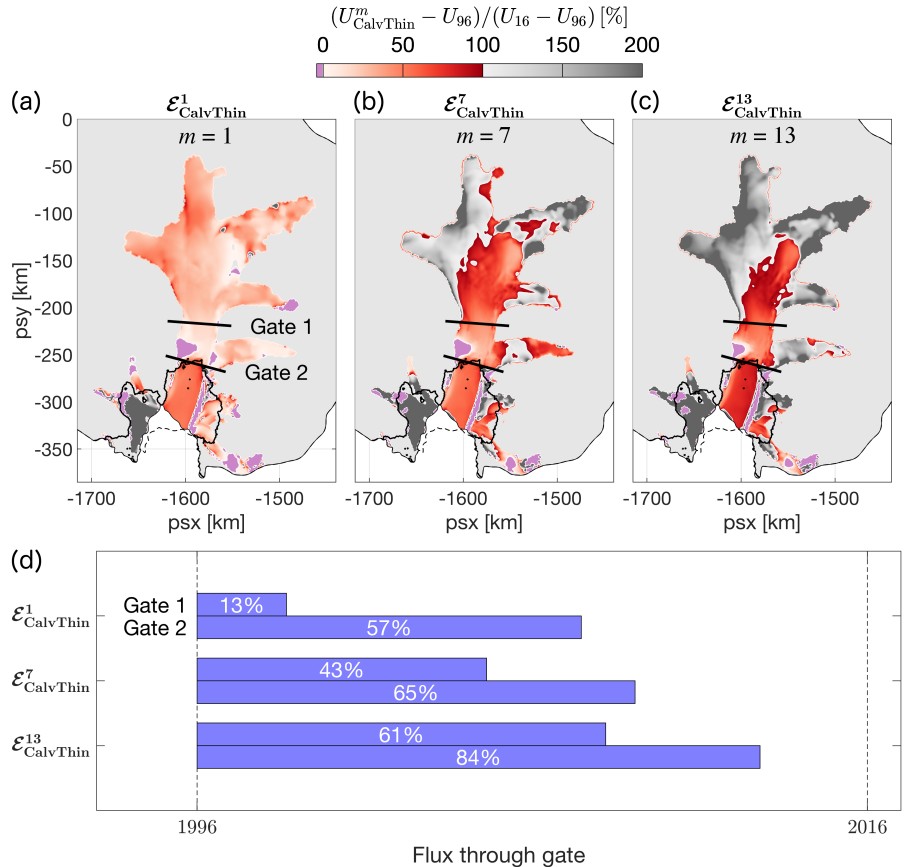

**Figure 5.** Dependency of simulated-versus-observed changes in surface speed on the sliding law exponent: **(a)** $m = 1$, **(b)** $m = 7$ and **(c)** $m = 13$. Larger values of $m$ cause an increased response of the modeled surface speed to geometrical changes (calving, thinning and grounding line retreat). For $m > 3$, the modeled response of slow-flowing ice in the upstream catchment exceeds observed changes by more than 2-fold, whereas for $m = 13$, modeled changes of the fast-flowing central trunk are still smaller than observed changes. **(d)** Changes in flux through Gate 1 and 2 as a percentage of observed changes for $m = 1, 7$ and 13.

at increments of two. Results for $m = 1, 7$ and 13 are shown in Fig. 5. A linear rheology induces a simulated response to calving and thinning that explains less than 50% of the observed changes everywhere. For $m = 7$, relative changes in flow speed exceed 100% along significant portions of the slower-flowing tributaries. For $m = 13$, which effectively corresponds to a plastic rheology, the modeled response overshoots observations by more than 100% in most areas, except along the main glacier, where the response approaches 100%. Across the model domain, a significant positive correlation exists between $m$ and relative velocity changes, indicating a stronger dynamic response to perturbations in geometry with increasing values of $m$. This finding is in agreement with Gillet-Chaulet et al. (2016) and Joughin et al. (2019), however our maps show that no single, spatially uniform value of the sliding exponent is able to produce a good match between model output and observations across the entire catchment.





The positive correlation between the flow response and $m$ is an inherent property of the adopted physical description of glacier dynamics. For the shallow ice stream approximation with a non-linear Weertman sliding law, the first-order response of the surface velocity, $\delta U$, to small perturbations in surface elevation, $\delta S$, was previously determined by Gudmundsson (2008)

and depends on $m$ in the following non-linear way :

$$\delta U \equiv |\mathcal{T}_{US}(m)|\delta S = \frac{f_1 m}{m + f_2}\delta S. \tag{5}$$

The transfer amplitude $|\mathcal{T}_{US}|$ contains complicated positive functions $f_1$ and $f_2$ that generally depend on the wavelength of the surface perturbation, geometrical factors such as the local bed slope, and the basal slipperiness $C$. Further details are provided in App. C. Despite the simplifying assumptions that underlie the analytical expression of $|\mathcal{T}_{US}|$ obtained by Gudmundsson

(2008), results from our simulations $\mathcal{E}^{m_i}_{\text{CalvThin}}, m_i \in \{1, 3, \cdots, 21\}$, indicate that Eq. 5 is also applicable to the more complex setting of PIG. Indeed, as explained in detail in App. C, we found that across a large portion of the PIG catchment, the transfer amplitude $|\mathcal{T}_{US}|$ provides a suitable model to describe the dependency of the relative velocity changes $\Delta U_{\text{CalvThin}}/\Delta U$ on $m$. The parameters $f_1$ and $f_2$ were treated as spatially variable fields, and best estimates for $f_1(\mathbf{x})$ and $f_2(\mathbf{x})$ were obtained by minimizing the misfit between $\frac{f_1(\mathbf{x})m}{m + f_2(\mathbf{x})}$ and $\frac{\Delta U^{m_i}_{\text{CalvThin}}}{\Delta U}(\mathbf{x})$ with $m_i \in \{1, 3, \cdots, 21\}$.

Given the non-linear dependency of $\Delta U_{\text{CalvThin}}/\Delta U$ on $m$ with known fields $f_1(\mathbf{x})$ and $f_2(\mathbf{x})$, one can derive an 'optimal' spatial distribution of the sliding exponent, $m_{\text{optimal}}(\mathbf{x})$, such that $\Delta U_{\text{CalvThin}}/\Delta U = 100\%$ everywhere, namely

$$m_{\text{optimal}}(\mathbf{x}) = \frac{100 f_2(\mathbf{x})}{f_1(\mathbf{x}) - 100}. \tag{6}$$

By construction, the variable sliding exponent $m_{\text{optimal}}(\mathbf{x})$ enables to reproduce 100% of the observed speed-up of PIG in response to calving and ice thickness changes. The results, depicted in Fig. 6a, indicate that plastic bed conditions ($m \gg 1$)

prevail across most of the fast-flowing central valley and parts of the upstream tributaries. Values generally increase towards the grounding line, whilst linear or weakly non-linear bed conditions are consistently found in the slow-flowing inter-tributary areas. This finding is compatible with the presence of a weak, water saturated till beneath fast-flowing areas of PIG, and hard bedrock or consolidated till between tributaries (Joughin et al., 2009).

Two interesting properties of the regression model in Eq. 5 are worth noting. Firstly, for $m \to \infty$, the function $|\mathcal{T}_{US}|$ ap-

proaches a horizontal asymptote with limit equal to $f_1$. As a consequence, the associated solution for $m_{\text{optimal}}$ diverges to $\infty$ for locations $x$ where $f_1(x) = 100$, and becomes negative where $f_1(x) < 100$. In these areas, indicated by black dots in Fig. 6a, no non-negative, finite value of $m$ exists such that $\Delta U_{\text{CalvThin}}(x)/\Delta U(x) = 100\%$, and conventional Weertman sliding is unable to fully reproduce the observed flow changes in response to thickness changes and calving. Either a different form of the sliding law is required, or additional changes in the rate factor $A$ and/or basal slipperiness $C$ are needed. These findings are the

subject of a forthcoming study. Our second observation concerns locations where $\Delta U$ either contains significant measurement uncertainties, or approaches the limit $\Delta U \to 0$. In these areas, the non-linear regression was generally found to be poor, with $R^2$ values smaller than 0.9 as indicated by the white dots in Fig. 6a. As no reliable estimate for $m_{\text{optimal}}$ could be obtained for areas shaded in white or black in Fig. 6a, values were instead based on a nearest-neighbour interpolation.

It is important to reiterate that the used regression method crucially relies on non-trivial measurements of changes in surface

velocity ($\Delta U \neq 0$), and cannot be used to retrieve information about the basal rheology of ice bodies that are presently in

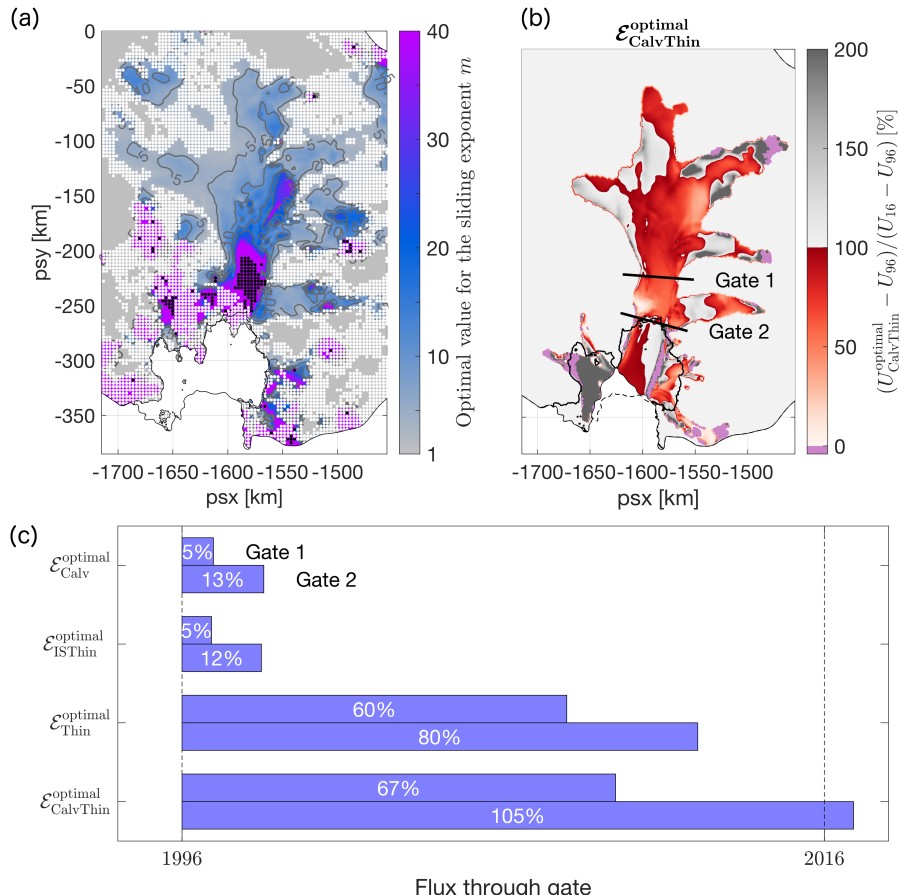

**Figure 6. (a)** Optimal values of the sliding exponent, required to ensure close agreement between modeled and observed changes in flow velocity of Pine Island Glacier between years 1996 and 2016. White and black dots mark areas where such an agreement cannot be achieved for different reasons: white dots indicate a poor fit between the transfer function $|\mathcal{T}_{US}|$ and $\Delta U_{\mathrm{CalvThin}}^{m_i}/\Delta U$, $m_i \in \{1, 3, \cdots, 21\}$, with $R^2 < 0.9$; black dots indicate areas where a positive, finite solution for $m_{\mathrm{optimal}}$ in Eq. 6 does not exists, and Weertman sliding cannot reproduce observed changes in surface flow. **(b)** Same as Fig. 3d but for optimal values of the sliding law exponent in panel **a**. **(c)** Same as Fig. 3e but for optimal values of the sliding law exponent in panel **a**.

steady state. It should also be noted that values of $f_1(\mathbf{x})$ and $f_2(\mathbf{x})$ were derived independently for each node of the computational mesh, whereas the continuum mechanical properties of glacier flow would suggest a non-zero spatial covariance $\langle f_1(x_1), f_1(x_2)\rangle \neq 0$ and $\langle f_2(x_1), f_2(x_2)\rangle \neq 0$. The optimal solution for $m$ is therefore not automatically mesh independent or robust with respect to the amount of regularization in the inversion. This concern is discussed further in App. D.

In order to demonstrate the improved model response to thinning and calving for a spatially variable sliding exponent $m_{\mathrm{optimal}}(\mathbf{x})$, we performed a new inversion with $m_{\mathrm{optimal}}(\mathbf{x})$, and subsequently repeated the geometric perturbation experiments $\mathcal{E}_*^{\mathrm{optimal}}$. The results are presented in Fig. 6b and c. Compared to spatially uniform values of $m$ (Fig. 3d and Fig. 5),





a spatially variable basal rheology generally improves the fit between observed changes in flow and the modeled response across the entire basin. Based on the flux changes through Gate 1 and 2, we find that (1) calving and ice thickness changes in

combination with a spatially variable, predominantly plastic bed rheology account for 67% and 105% of flux changes through Gate 1 and 2 respectively, compared to 28% and 64% for a uniform viscous sliding law with exponent $m = 3$, that (2) calving and ice shelf thinning caused an almost identical response in ice dynamics upstream of the grounding line, and that (3) dynamic thinning and grounding line movement account for most of the flux changes between years 1996 and 2016. The remaining mismatch between the observed and modeled response in Fig. 6b can, at least in part, be attributed to uncertainties in $m_{\mathrm{optimal}}(\mathbf{x})$.

This is of particular relevance in the vicinity of the grounding line and for parts of the central trunk, where the non-linear regression method in Eq. 5 did not provide a reliable or finite estimate for $m_{\mathrm{optimal}}$, and where Weertman theory of sliding could break down all together (Iverson et al., 1998; Schoof, 2006).

## 4 Conclusions

Based on the most comprehensive observations of ice shelf and grounded ice thickness changes to date, and a suite of di-
agnostic model experiments with the contemporary flow model Úa, we have analyzed the relative importance of ice shelf thinning, calving and grounding line retreat for the speed-up of Pine Island Glacier between years 1996 and 2016. The detailed comparison between simulated and observed changes in flow speed has provided unprecedented insights into the ability of a modern-day ice flow model to reproduce dynamic changes in response to prescribed geometric perturbations. Significant discrepancies between observed and modeled changes in flow were found, and were addressed by either allowing changes in
ice viscosity and basal slipperiness, or by varying the mechanical properties of the ice-bed interface. For viscous sliding at the bed, geometric perturbations could only account for 64% of the observed flux increases close to the grounding line, whereas the remaining 36% could be attributed to large and widespread changes in ice viscosity (including damage) and/or changes in basal slipperiness. Under the alternative assumption that ice viscosity and basal slipperiness did not change considerably over the last two decades, we found that the recent increase in flow speed of Pine Island Glacier is only compatible with observed
patterns of thinning if a heterogeneous, predominantly plastic bed underlies large parts of the central glacier and its upstream tributaries.

*Code and data availability.* The open-source ice flow model Úa is available from Gudmundsson (2020). All model configurations files specific to this study, as well as model output and plotting routines for each figure are available from DOI TBC. Ice shelf thinning rates are available upon request from FP (fernando.serrano.paolo@jpl.nasa.gov).

**Appendix A: Observations of Pine Island Ice Shelf thickness changes between 1996 and 2016**

We derived a new ice-shelf height time series from measurements acquired by four overlapping ESA satellite radar altimetry (RA) missions: ERS-1 (1991–1996), ERS-2 (1995–2003), Envisat (2002–2012), and CryoSat-2 (2010–present). For this





study, we constructed a record of ice-shelf height spanning 20 years (1996–2016), with a temporal sampling of 3 months. We integrated all measurements along the satellite ground tracks and gridded the solution on a 3 by 3 km grid.

Our adopted processing steps for RA data are a modification/improvement from Paolo et al. (2016) and Nilsson et al. (2016). Specifically for CryoSat-2, we retracked ESA's SARIn L1B product over the Antarctic ice shelves using the approach by Nilsson et al. (2016); we corrected for a 60 m range offset for data with surface types 'land' or 'closed sea'; and removed points with anomalous backscatter values (>30 dB). We estimated heights with a modified (from McMillan et al. (2014)) surface-fit approach, with a variable rather than constant search radius to account for the RA heterogeneous spatial distribution,

and calculating mean values along the satellite reference tracks; we removed height estimates less than 2 m above the Eigen-6C4 geoid (Chuter and Bamber, 2015) to account for ice-shelf mask imperfections near the calving front; applied all of the standard corrections to altimeter data over ice shelves (for example, removed gross outliers, and residual heights with respect to mean topography > 15 m; ran an iterative three-sigma filter; minimized the effect of variations in backscatter (Paolo et al., 2016); corrected for ocean tides (Padman et al., 2002) and inverse barometer effects (Padman et al., 2004).

We then gridded the height data in space and time on a $3\,\mathrm{km} \times 3\,\mathrm{km} \times 3\,\mathrm{month}$ cube, for each mission independently. We merged the records (all four satellites) by only accepting time series that overlapped by at least three quarters of a year to ensure proper cross-calibration, and removed (and subsequently interpolated) anomalous data points that deviated from the trend by more than 5 std. This removes data with, for example, satellite mispointing, anomalous backscatter fluctuations, grounded-ice contamination, high surface slopes and geolocation errors. We fitted linear trends to the gridded product to obtain the $\Delta H$

field used in our model experiments (see Sect. 2.1). We also removed a 3 km buffer around the ice-shelf boundaries to further mitigate floating-grounded mask imperfections, and the limitation of geophysical corrections within the ice-shelf flexural zone.

**Appendix B: Model configuration and inverse methodology**

The open source ice flow model Úa (Gudmundsson, 2020) uses finite element methods to solve the shallow ice stream equations, commonly referred to as SSA or SSTREAM, on an irregular triangular mesh. The diagnostic velocity solver is based on

an iterative Newton-Raphson method. A fixed mesh with 109,300 linear elements was used with a median nodal spacing of 1.2 km and local mesh refinement down to 500 m in areas with above-average horizontal shear, strong gradients in ice thickness and within a 10 km buffer around the grounding line. The mesh was generated using the open-source generator mesh2d (Engwirda, 2014).

The inverse capabilities of Úa follow commonly applied techniques in ice flow modeling to optimize uncertain model

parameters, $p_i$, based on prior information, $\hat{p}_i$, and a range of observations with associated measurement errors. Úa uses an adjoint method to obtain a combined optimal estimate of the spatially varying rate factor $A$ and basal slipperiness $C$ across the full model domain, for given observations of surface velocity $u_{\mathrm{obs}}$ and measurement errors $\varepsilon_u$. Optimal values for $p_i \in \{A, C\}$ were obtained as a solution to the minimization problem $d_p J$ with the cost function $J$ defined as the sum of the misfit term $I$





and Tikhonov regularization $R$: $J = I + R$, with

$$I = \frac{1}{2\mathcal{A}} \int d\mathbf{x} \left(u_{\mathrm{model}} - u_{\mathrm{obs}}\right)^2 / \varepsilon_u^2 \,, \tag{B1}$$

$$R = \frac{1}{2\mathcal{A}} \int d\mathbf{x} \sum_i \left( \gamma_s^2 \left( \nabla \log_{10}(p_i/\hat{p}_i) \right)^2 + \left( \log_{10}(p_i/\hat{p}_i) \right)^2 \right) \,, \tag{B2}$$

and $\mathcal{A} = \int d\mathbf{x}$ the total area of the model domain. A priori values of the rate factor and slipperiness were chosen as $\hat{A} = 5.04 \times 10^{-9}\,\mathrm{kPa}^{-3}\,\mathrm{yr}^{-1}$, which corresponds to a spatially uniform ice temperature of -15 °C (Cuffey and Paterson, 2010), and $\hat{C} = u_b \tau^{-m}$, with $u_b = 750\,\mathrm{m\,yr}^{-1}$ and $\tau = 80\,\mathrm{kPa}$ and $m$ the sliding law exponent. An iterative interior point optimization algorithm was used to calculate $d_p J$ and stopped after $10^4$ iterations, when fractional changes to the cost function were less than $10^{-5}$. An optimal value for the Tikhonov regularization multiplier, $\gamma_s$, in the cost function was determined using an L-curve approach, as shown in Fig. B1. The value $\gamma_s = 25000$ m was used for all experiments in the main part of the text, as it produced the smallest misfit between observed and modelled surface velocities whilst limiting the risk of overfitting. The sensitivity of the main results with respect to the choice of $\gamma_s$ is discussed in App. D.

Figure B1 shows the difference between $u_{\mathrm{model}}$ and $u_{\mathrm{obs}}$ (panel **b**), and corresponding optimal estimates of $A$ (panel **c**) and $C$ (panel **d**) for $\gamma_s = 25000\,\mathrm{m}$. Modeled surface velocities are typically within 30 meters per year or less of the observed values, with a mean misfit of $-1.68\,\mathrm{m\,yr}^{-1}$ and standard deviation of $15.3\,\mathrm{m\,yr}^{-1}$. The highest values of the rate factor are generally found within the shear margins, with positive equivalent ice temperatures suggesting the presence of a complex rheology or damage. The highest values of the slipperiness are consistently found in the fast-flowing central part of the glacier and along its upstream tributaries, with noticeably reduced values of $C$ in an area between 5 and 40 km upstream of the 1996 grounding line. These results are broadly in agreement with previously published maps, see e.g. Arthern et al. (2015).

## Appendix C:  Non-linear dependency of the flow response on the sliding exponent

The transfer amplitude $|\mathcal{T}_{US}|$, defined in Eq. 5, describes the linear response of the along-slope surface velocity to small harmonic perturbations in the surface elevation or, equivalently, ice thickness. Analytical solutions for the transfer function $\mathcal{T}_{US}$ (amplitude and phase) in the framework of the shallow ice stream approximation with a linear ice rheology ($n = 1$ in Eq. 3) and non-linear viscous Weertman sliding (arbitrary $m$ in Eq. 4) were previously obtained by Gudmundsson (2008). Note that the original expression (Eq. 29 in Gudmundsson (2008)) contained a printing error so we repeat the correct form here:

$$\mathcal{T}_{US} = \frac{\tau_d \left[ m\gamma(1+\psi) + \eta H \left(j^2\psi + k^2 + 4l^2\right) \right]}{Hm\gamma^2 + \gamma\eta H^2 \left[l^2(4+m) + k^2(1+4m)\right] + 4H^3 j^4 \eta^2} \,, \tag{C1}$$

where $H$ is the local ice thickness, $\alpha$ is the local bed slope, $\rho$ is the ice viscosity, $\tau_d = \rho g H \sin\alpha$ is the driving stress, $\eta$ is the effective viscosity and $\gamma = \frac{\tau_d^{1-m}}{mC}$, $\psi = \mathrm{i}kH\cot\alpha$ and $j^2 = k^2 + l^2$ are abbreviations, with $k$ and $l$ the along-slope and transverse wavelength respectively of the harmonic surface perturbation. Since we focus on the instantaneous response of the velocity to perturbations at the surface, the exponential decay of $\mathcal{T}_{US}$ with time has been omitted. An equivalent expression for the response of the transverse velocity component can be derived; we refer to Gudmundsson (2008) for more details.

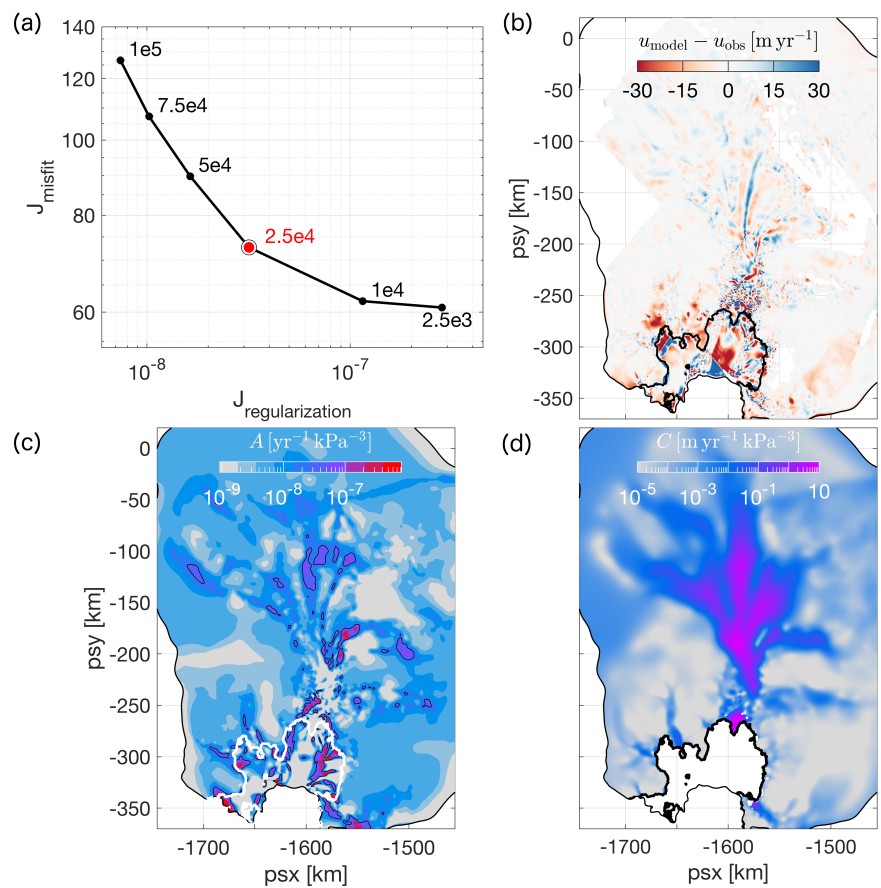

**Figure B1.** (a) L-curve used to determine the optimal value of the Tikhonov regularization multiplier $\gamma_s$, highlighted in red. (b) Misfit between modeled and observed surface speed in 1996 for $\gamma_s = 25000\,\mathrm{m}$. (c) Rate factor ($A$ in Eq. 3) in 1996, obtained as a minimum of the cost function $J$ in Eq. B1 with $\gamma_s = 25000\,\mathrm{m}$. The equivalent depth-averaged ice temperature ranges from -35 °C (grey) to 5 °C (red). Colors are discretized at 5 °C intervals and the black lines indicate the 0 °C contour. The white line corresponds to the 1996 grounding line position. (d) Optimal value of the basal slipperiness ($C$ in Eq. 4) in 1996, estimated using the adjoint minimization approach.

Following Gudmundsson (2008), physical quantities can be rescaled to obtain the non-dimensional form of the transfer function. After substitution of the scalings $H \rightarrow 1$, $\eta \rightarrow 1/2$, $\tau_d \rightarrow 1$ into Eq. C1 and some reordering, one obtains

$$\mathcal{T}_{US} = \frac{m\left[\frac{1}{C}\left(1+\psi\right) + \frac{1}{2}\left(j^2\psi + k^2 + 4l^2\right)\right]}{m\left[j^4 + \frac{1}{2C}\left(l^2 + 4k^2\right)\right] + \frac{1}{C^2} + \frac{1}{2C}\left(4l^2 + k^2\right)}. \tag{C2}$$

The resulting transfer amplitude takes the form $|\mathcal{T}_{US}| = \frac{f_1 m}{m + f_2}$ as in Eq. 5, where functions $f_1$ and $f_2$ depend on $C$, $\alpha$, $k$ and $l$.

        The analytical expression in Eq. C2 describes the first-order response to small perturbations in ice thickness, $\delta H \ll 1$, for
well-defined length scales characterized by $k$ and $l$. However, in a realistic setting such as PIG, the system responds to a complicated perturbation composed of a range of wavelengths and amplitudes, and Eq. C2 does not automatically hold. Based



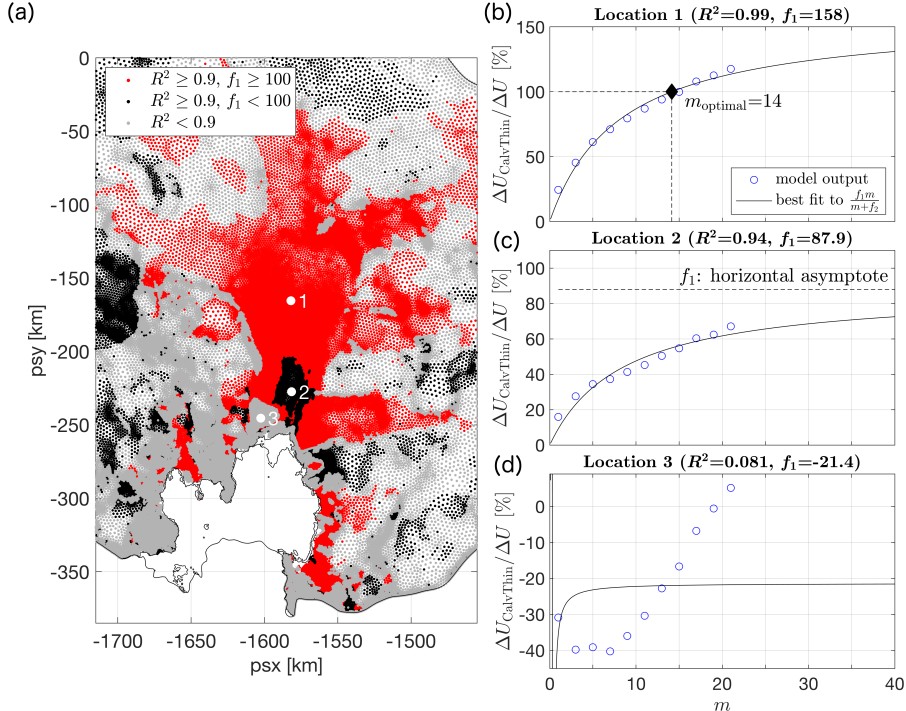

**Figure C1.** **(a)** Goodness of fit between $\frac{f_1 m}{m+f_2}$ and model simulations $\Delta U^{m_i}_{\mathrm{CalvThin}}/\Delta U$, $m_i \in \{1,3,\cdots,21\}$. Red areas correspond to $R^2 \geq 0.9$ and fitting parameter $f_1 \geq 100$. An example of the fit at location 1 and resulting $m_{\mathrm{optimal}}$ (Eq. 6) are shown in panel **b**. Black areas in **(a)** correspond to $R^2 \geq 0.9$ and fitting parameter $f_1 < 100$. The horizontal asymptote with limit $< 100$ indicates that a positive, finite solution $m_{\mathrm{optimal}}$ does not exist, and Weertman sliding cannot reproduce 100% of the observed changes in surface velocity. An example of the fit and asymptote at location 2 are shown in panel **c**. Grey areas in **(a)** correspond to $R^2 < 0.9$, indicating a poor fit between $\frac{f_1 m}{m+f_2}$ and $\Delta U^{m_i}_{\mathrm{CalvThin}}/\Delta U$, $m_i \in \{1,3,\cdots,21\}$. An example at location 3 is shown in panel **d**.

on experiments $\mathcal{E}^{m_i}_{\mathrm{CalvThin}}$, $m_i \in \{1,3,\cdots,21\}$ presented in Sect. 3.3, we found that the simulated surface response of PIG to observed geometrical perturbations retains it dependency on $m$ of the form $\frac{f_1 m}{m+f_2}$, but more complicated expressions for $f_1$ and $f_2$ are required that do not exist in analytical form. A best-estimate for the spatially varying fields $f_1$ and $f_2$ was obtained by

minimizing the misfit between $\Delta U^{m_i}_{\mathrm{CalvThin}}/\Delta U$, $m_i \in \{1,3,\cdots,21\}$ and $\frac{f_1 m}{m+f_2}$. The resulting misfit, quantified by $R^2$ values, is summarized in Fig. C1a. Red and black areas indicate a good fit with $R^2 \geq 0.9$, though an important distinction was made between solutions with $f_1 \geq 100$ (red) and $f_1 < 100$ (black). The difference between both cases is explained further in Sect. 3.3. Examples of the fit at locations 1 and 2 are shown in Fig. C1b and c respectively. Grey shading in Fig. C1a corresponds to a poor fit ($R^2 < 0.9$) and the dependency of $\Delta U^m_{\mathrm{CalvThin}}/\Delta U$ on $m$ cannot be adequately described by the function $\frac{f_1 m}{m+f_2}$.

Possible reasons for this discrepancy are discussed in Sect. 3.3.



## Appendix D: Dependency of the results on the regularization

The inverse problem of inferring information about the rate factor $A$ and basal slipperiness $C$ from uncertain observations of surface velocity is generally ill-posed. To remedy the ill-posedness of the problem, additional information in the form of a regularization term (Eq. B2) is commonly added to the cost function. In a Bayesian framework, the regularization plays the role

of a prior and is added to the misfit, which corresponds to the likelihood. The solution of the minimization problem generally depends on the choice of regularization. In the specific case of a Tikhonov regularization, which is used throughout this study, the solution for $A$ and $C$ will depend on the unknown multiplier $\gamma_s$ in Eq. B2. One method to choose an 'optimal' value for $\gamma_s$ is the L-curve approach presented in App. B. However, this is an ad-hoc method and it remains to be shown that results are robust for a range of $\gamma_s$ values.

In case of the perturbation experiments $\mathcal{E}_*^3$, which were designed to simulate the velocity response to a series of prescribed changes in the PIG geometry, we are primarily interested in the $\gamma_s$-dependency of the relative fluxes in Fig. 3e. In addition to the experiments with default value $\gamma_s = 25000\,\mathrm{m}$, identical perturbation experiments were carried out for $\gamma_s = 10000\,\mathrm{m}$ and $\gamma_s = 50000\,\mathrm{m}$. The corresponding changes in flux, presented in Table D1, do not show any significant variability with $\gamma_s$ and results presented in Sect. 3.1 can be considered robust, at least across the range of tested $\gamma_s$ values.

Experiments $\mathcal{E}_A^3$ and $\mathcal{E}_C^3$ were also repeated for $\gamma_s = 10000\,\mathrm{m}$ and $\gamma_s = 50000\,\mathrm{m}$. Maps of $A$ and $C$ (not shown) were compared to the default results for $\gamma_s = 25000\,\mathrm{m}$ shown in Fig. 4, and no significant qualitative differences were found.

Perturbation experiments $\mathcal{E}_*^m$ for a range of sliding law exponents $1 \leq m \leq 21$ were repeated for $\gamma_s = 10000\,\mathrm{m}$ and $\gamma_s = 50000\,\mathrm{m}$. Following the approach outlined in Sect. 3.3, an optimal spatial distribution of the sliding exponent was computed for each $\gamma_s$. Results are presented in Fig. D1 and show a decreasing trend in $m_{\mathrm{optimal}}$ for increasing values of the regularization

multiplier $\gamma_s$. In particular, the area where no positive, finite solution exist for $m_{\mathrm{optimal}}$ (shaded in black) is reduced in size and eventually disappears for increasing amounts of regularization. However, the spatial distribution of $m_{\mathrm{optimal}}$ is found to be in broad agreement across the considered range of $\gamma_s$.

*Author contributions.* JDR and RR designed and initiated the project and prepared the manuscript; FP processed the ice shelf thickness data; JDR performed the model simulations, carried out the analysis and produced the figures; FP and GHG reviewed and edited the paper.

*Competing interests.* JDR serves as topical editor for The Cryosphere.

*Acknowledgements.* JDR, GHG and RR are supported by the TiPACCs project that receives funding from the European Union's Horizon 2020 research and innovation programme under grant agreement no. 820575. RR is further supported by the Deutsche Forschungsgemeinschaft (DFG) by grant WI4556/3-1 and GHG by the NSFPLR-NERC grant: *Processes, drivers, predictions: Modelling the response of Thwaites Glacier over the next century using ice/ocean coupled models* (NE/S006745/1).





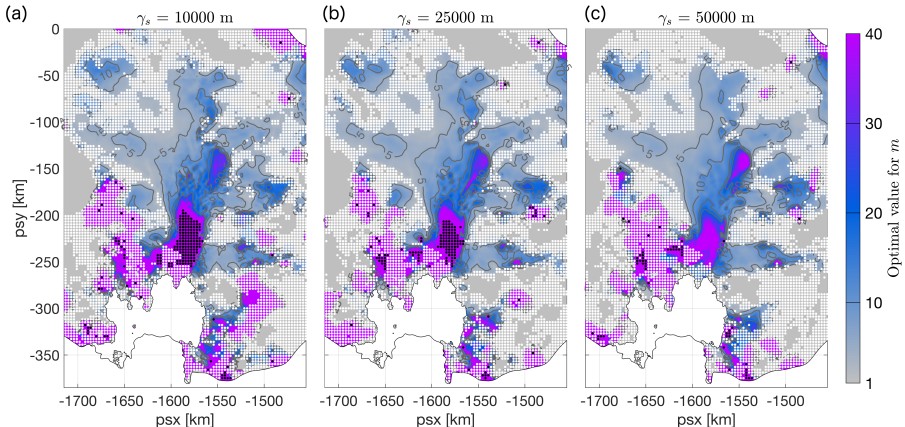

**Figure D1.** Optimal distribution of $m$, as in Fig. 6a, for different values of the regularization multiplier: **(a)** $\gamma_s = 10000\,\text{m}$, **(b)** $\gamma_s = 25000\,\text{m}$ and **(c)** $\gamma_s = 50000\,\text{m}$. White dots indicate areas where results for the non-linear regression method were poor, with a $R^2$-value smaller than 0.9. Black dots indicate areas where the value of $f_1$ in the fit is less than 100, indicating that agreement between simulated and observed changes in surface velocity cannot be achieve for finite values of $m$. The value $\gamma_s = 25000$ m was used throughout the main part of this study.

**Table D1.** Sensitivity of the relative flux changes in the $\mathcal{E}_*^3$ experiments (see Fig. 3) with respect to the choice of regularization multiplier $\gamma_s$. The optimal value, $\gamma_s = 25000$ m, used throughout this study was based on the L-curve presented in Fig. B1.

|  |  | $\gamma_s = 10000\,\text{m}$ | $\gamma_s = 25000\,\text{m}$ | $\gamma_s = 50000\,\text{m}$ |
|---|---|---|---|---|
| $\mathcal{E}_{\text{Calv}}^3$ | Gate 1 | 2% | 2% | 2% |
|  | Gate 2 | 15% | 13% | 13% |
| $\mathcal{E}_{\text{ISThin}}^3$ | Gate 1 | 2% | 2% | 2% |
|  | Gate 2 | 14% | 13% | 12% |
| $\mathcal{E}_{\text{Thin}}^3$ | Gate 1 | 24% | 26% | 25% |
|  | Gate 2 | 38% | 45% | 42% |
| $\mathcal{E}_{\text{CalvThin}}^3$ | Gate 1 | 26% | 28% | 27% |
|  | Gate 2 | 58% | 64% | 58% |

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
