# Peer review of "Drivers of Pine Island Glacier speed-up between 1996 and 2016"

_The Cryosphere, 2020_

## Referee Comment (RC1) · Anonymous Referee #1 · 22 Jul 2020

Overview This is a nice paper showing some well-constructed model experiments designed to understand the sensitivity of PIG to various parameter. Despite my many comments and criticisms, I would very much like to see this paper published and I think it makes a valuable contribution to the field.

General Comments

I would like to see more discussion on the quality of the data and how it affects the results. To me the quality of the 1996 data from radar altimetry are highly suspect – ERS DEMS can be off by 10s or 100s of meters in places. Figure 1c does nothing to improve my confidence in these data, particularly the strong gradient in the thickness change across the ice shelf, with nearly zero thinning on the shelf centerline. Thus, I expect some model discrepancies could be explained by errors in the data sets used

to constrain the model. Related to this issue, I would rather see results like Figure 4b expressed as changes in basal shear stress. Particularly in light of the noisy 1996 data set, I would expect inferred changes in basal shear stress to be largely due to topographic (driving stress) errors.

As mentioned in some of the comments below, I have concerns about the joint inversion for A and C. While some prior info seems to be used, it's not well documented in the appendix. For example, do the pˆ priors allow greater variation of A in the margins, while suppressing it elsewhere. I suspect not from the outputs. The inversions for A bear no resemblance to what my intuition would say the distribution should look like. There are numerous papers debating factors of 10 enhancement in shear margins, but this solution as irregular, patchy variations that vary by more than a factor of 100 (shouldn't the priors have not let A vary so much), sometimes with blobs of effectively very cold/stiff ice crossing the margins. I would really have liked the results better if the inversions for A had been restricted to the shelf where they don't make an already ill-posed problem even more so. This would have allowed for better discrimination of any role damage (A enhancement) on the ice shelf shear margins.

Other papers have presented similar results using a forward approach with somewhat better agreement with the data, though for a shorter time period (e.g., Gillet-Chaulet et al 2016 and Joughin et al 2019). There is some discussion about this work, but more comparisons could be drawn. For example, to get good agreement by assuming a reduction in basal traction near the grounding line driving by height-above flotation (effective pressure) variations. Could something like that be done here.

Figures. The colour maps are all in shades of red are hard to interpret. Maybe a different colour map or 25% contours would help.

Specific Comments, technical corrections, minor wordsmithing

Line 27: Would be fair to cite Seroussi et al 2014 (doi:10.5194/tc-8-1699-2014) and Joughin et al, 2010 (in refs) in this list. Alternatively, these might be better in line 34.

Line 70-75. Where does loss of traction due grounding line fit it, which is caused by thinning (this is probably a bigger factor than the original loss of buttressing that triggered it – e.g., Figure 4 Joughin 2019 ref, which thinning alone can actually slow ice shelf a GL velocities without the loss of traction from ungrounding). Payne et al 2004 also look at the shift of the GL, rather than just the loss of ice shelf thinning.

Line 84 deltaUA+deltaUC=0. Again is the loss of traction as the grounding line retreats being bookkept as a change in deltaUC or a change deltaUthin. This is also important for interpreting line 95 because the paper cited shows the thinning driven response, but that includes loss of traction as the ice approaches flotation and finally ungrounds. All I am asking here for is a sentence clearly stating where the loss of traction due to ungrounding is bookkept.

Line 139. "The resulting values for $\Delta H$, linearly interpolated across the grounding line and in data sparse areas, are shown in Fig. 1c" I don't see how this is valid or maybe I am not understanding what is being done. You could have a case where the shelf thins by 10s or 100s of meters due to melting, but the grounded change in thickness is far less. Am I misinterpreting something.

Line 165. "linear, viscous or close-to plastic" It should be "linear viscous, non-linear viscous, or close to plastic" Line 167: "which caused small variations in $\tau$b between cases, " This has to do with the interplay between A and taub, not so much the exponent. If you had used a fixed A, then taub is in fact what you are using to achieve stress balance with its parameterization via the sliding law (any sliding law). So, I am not sure the following sentence is correct and probably should be removed. What is different about the two studies is that A is determined on the entire domain for this study vs only on the shelf. A such additional degrees of freedom are introduced, which will give a better fit to the data, but could result in model parameters that are far from the true physicl values (eg. To stiff A balanced by a too slippery bed). Given that the response is dependent on the flow and sliding law parameters, the accuracy of the fit is somewhat immaterial. There is a counter argument that many other studies fit for one parameter

or another using an assumed value for the other, so there could be errors in the assumed value. But I am not seeing a strong case for the improvement here. Line 172: I may be playing with semantics here but what is obtain is not a "best estimate" it's an estimate that produces the best fit, but as mentioned above, this by no means ensures that it is the best estimate. Line 176 "basal slipperiness were kept fixed" Please how the ungrounded region is treated – either here or above. While it may seem obvious, when various parameters are being held fixed, it's not always immediately clear that the traction is being zeroed when the ice goes afloat. Line 4 51: Why not initialize with an A estimated from temperature model; solutions are often better if you start nearer the answer. Also given that there are blobs of A in close proximity each other that can vary by a couple of orders of magnitude (a difference between ice at 0 and -30 deg) without any real correspondence to the flow pattern (outside of some shear margins) it would be worth some comment here about how physically realistic the solution is at least in a qualitative sense. Even for shear margins, the lower trunk on the west side where you would expect some enhancement, the ice actually looks like it's much stiffer than the reference -15 deg.

Line 216: Would be nice to have a brief introduction to this section between 3 and 3.1. Actually this would be a good place to describe the flux gates, which break up the flow of the text below. Line 221: "viscous, rate-strengthening bed rheology " Please add "non-linear" viscous here. Also please note that this description applies under the assumption of till at the ice bed interface. As conceived, the original Weertman law applies to sliding over a hard, non-deforming bed, with ice and rock separated by a thin water film. Inversions here and elsewhere suggest both types of bed may be present beneath PIG. To the extent that its applicable to till, its likely only at low sliding speeds (see Zoet reference). Line 224-225: 50% speedup. Be clear on where, it looks like this magnitude really applies only to the outer shelf. Might also be good to note that results are consistent with the Schmeltz et al 2002 reference that is cited in the text, where a similar experiment was performed (they only get a speedup of $\sim$40% right at the shelf front – mid shelf its more like 20%). Would be far more accurate to say "restricted

to the OUTER ice shelf". A lack of speed from calving at the GL is consistent with a number of published results showing speed on PIG over the last decade, during which time there have been large calving events. Line 232-237.5: Please consider moving this flux gate description to intro paragraph before 3.1 as described above. Line 237: "This supports..." This what? How about something like "This moderate response supports..." Now please cite the earlier works that your work supports. Lines 239-243: This text could be wordsmithed – there are least 3 place saying something about the similarity to calving response. Could just say something like "The responses to calving and thinning are similar at the flux gates, but the calving induces a larger response on the outer shelf (50% vs < 25%). "Ice shelf thinning is generally accepted to be the main driver of ongoing mass loss of PIG, and patterns of ice shelf thinning elsewhere in Antarctica are strongly correlated to observed changes in grounding line flux (Reese et al., 2018; Gudmundsson et al., 2019)." This is a gross over-simplification of the conventional wisdom, which is that loss of buttressing can induce near GL thinning, which leads to retreat, which induces more loss. As AR5 notes, "Problems arise at the GL because, in addition to flotation, basal traction is dramatically reduced as the ice loses contact with the underlying bedrock (Pattyn et al., 2006). This is a topic of active research, and a combination of more complete modelling of the GL stress regime (Favier et al., 2012) and the use of high-resolution (subkilometre) models (Durand et al., 2009; Cornford et al., 2013) shows promise towards resolving these problems." Numerous studies have shown that the near GL speedup is a response to the loss of traction as the GL retreats, not the thinning itself, although that thinning believe to have triggered the whole process, likely through smaller speedups that thin the GL to flotation. Line 246: "in upstream flow, CONSISTENT WITH OTHER STUDIES THAT HAVE SHOWN SIMILAR RESULTS [e.g., Payne 2004, Joughin 2010, 2019, Seroussi, 2014, Schoof 2007....]. Line 271 please add also "non-linear" before viscous. Also "non linear viscous bed rheology described by a Weertman sliding law". Weertman is not a viscous flow model even if the expressions are mathematically equivalent and generally is taken to mean an exponent between 2 and 3. Since your law applies to a broader

range of exponents and indeterminate bed conditions (probably both till and hard bed), how about just substituting here and above power-law sliding in place of Weertman sliding law. At some point where you used m=3 you can say "power-law sliding with m=3, which for hard beds corresponds to a Weertman sliding law" to credit Weertman (a reference to his work would be nice). Line 290 "Weertman sliding" change to "power law sliding" Line 300: Change "Weakening of the ice in these areas accounts for" to "Weakening of the ice in these areas is sufficient to account for"... Its only one model on the knob, so it's a sufficient but not necessary condition. Line 302-303: I agree this change is physically improbable. But as noted above, so is the original A, which effectively has very warm ice in the margin just upstream of the margin where the ice very cold. See other comments about this. Line 368: The result is also consistent with the Schoof type sliding law used by Joughin et 2019, which produces Coulomb plastic like behavior at speeds > 300 m/yr and low-m (Weertman-like) behavior at slower speeds. The areas with plastic like behavior in Figure 6 transition to lower exponents in the area between the 600 and 100 m/yr contours shown in Fig 4, which would be worth commenting on. Line 387: "Compared to spatially uniform values of m....". But this in effect what a sliding like that proposed by Schoof (and in another paper by Gagliardini et al) does – provides high-m behavior for fast flow and low-m behavior for slow flow. Line 399: Change "Based on the most comprehensive observations..." By what metric? Other data sets are arguably more comprehensive (you have only 2 snapshots in time). Whether they are or are not the most comprehensive has no bearing on the value of this paper. Instead, simply say "Based on a comprehensive..." and you are on firm ground. Line 402: Remove "unprecedented". The are some nice results in this paper, but they largely echo the results of earlier work (I mean no slight here this is true of most papers). Line 408: See general comments, but Joughin et al for a different time period got quite close agreement in a similar set of experiments by changes in basal shear stress proportional to the height above flotation in the region immediately above the grounding line. A compare and contrast sentence on this point would be good.

L410: we found that the recent increase in flow speed of Pine Island Glacier is only

compatible with observed patterns of thinning if a heterogeneous, predominantly plastic bed underlies large parts of the central glacier and its upstream tributaries, CONSISTENT WITH EARLIER FINDING (there are several that are appropriate).

---

## Referee Comment (RC2) · Stephen Cornford (Referee) · 17 Aug 2020

This paper attributes the speed up in Pine Island Glacier over a 20 year period to a combination of ice thinning, calving front change and rheological change. It concludes (as have others) that the PIG trunk appears to be subject to Coulomb sliding, at least in some parts. This is a well written paper based on sound methods. I do have some minor issues with the manuscript.

[Figure]

**General Comments**

L65 (eqn 2). The right hand side does not make sense to me for the reasons you point out later: you can't determine these things individually and then add them up. You can certainly (as you do) look at each in turn, and even talk about combinations. I can see why you want a notation like this, but at the end of the day, the equation is not correct given the usual definition of addition.

L165 : The inverse problem is solved for each $m$, and $T_b$ is slightly different. But that would not be so if the velocity were the same in each case, and the velocity could be the same in each case (in the optimal case, equal to the observations). The differences must be due to optimization method or similar. Joughin's method is equally valid, so you should not claim that yours is more appropriate.

**Specific Comments**

L30: "model studies have primarily focused on the important problem of simulating the response of PIG to a potential anthropogenic intensification of melt." Perhaps – but Joughin 2010 and 2019 were also concerned with attributing cause to observed change. This is mentioned later (from L42), so you have not neglected these papers, but it does make it makes 'primarily' incorrect

L95: Joughin et al. (2019) *and Joughin 2010*

2.2.1 Inverse experiments, and general use of inverse in a 'slang' fashion. These are optimization experiments, which involve solution of inverse problems. Ua does not have inverse capability, it has optimization capability.

Figure 3. Either have 'unaccounted' bars in every row, or none.

**L362.** $m_{optimal} = f_2/(f_1 - 1)$ ? $100\% \neq 100$.

L500 "In a Bayesian framework, the regularization plays the role of a prior and is added to the misfit, which corresponds to the likelihood". Tikhonov regularization can be seen as derived from Bayes rule with certain assumptions about the priors. But the methods used here are, I would say, not derived from Bayes rule because the prior is not determined from additional observation or theory: it is chosen to regularize the solution, and the relation to Bayes rule is incidental. Maybe a personal bugbear of mine.

---

## Referee Comment (RC3) · Anonymous Referee #3 · 22 Aug 2020

Review of "Drivers of Pine Island Glacier retreat from 1996 to 2016" by J. De Rydt et al.

The manuscript "Drivers of Pine Island Glacier retreat from 1996 to 2016" by J. De Rydt and colleagues analyses the role of several processes in causing the observed changes of Pine Island Glacier between 1996 and 2016 using numerical modeling. They estimate the relative role of calving, ice shelf thinning, ice shelf and ice sheet thinning, and well as combinations of these changes and their ability to reproduce the observed changes in ice velocity. The manuscript is well written, well explained, the figures are appropriate and this work is important as we try to better understand the glacier's changes in this region. There is one major point, however, that I would like to see addressed to make sure the results are robust and the conclusions not impacted

by model parameter.

The 1996 velocities are reproduced by inverting simultaneously the rigidity parameter A and the basal friction parameter C. This has been done in several studies, however there is an infinite combination of these two parameters that can yield similar velocity fields with drastically different values for each of the parameters. This is something that is observed for the inversion of 2016 in the present manuscript: two additional inversions are done to fit the velocity observations for that year, one changing only C, and the other one changing only A. Therefore, two different combinations of parameters (A1996 with C2016 and A2016 with C1996) yield similar velocity fields in agreement with the 2016 observations. This is a recurrent problem in our field and the alternatives are limited, however I am wondering if a different combination of parameters would lead to different conclusions and I would like to see this point addressed. One solution to do so could be for the 1996 observations to first optimize the friction parameter and then the rigidity parameter in a first case, while a second case would first optimize the rigidity parameter and the friction afterwards. I expect these two cases to exhibit significant differences as most of the misfit will be captured by the first model parameter inferred, with the second one only capturing "residual-like" misfit. Experiments done with these two initial conditions will allow to make sure that the conclusions are robust and not impacted by the choice of these parameters.

Major comments

As explained in detail above, I would like to make sure that the conclusions are robust under a range of values for A and C as both parameters are unconstrained. Additional experiments with other values such as proposed above would help ensure that this is the case.

The role of grounding line retreat is confusing. It is not part of ice shelf thinning, but it is included in the ice sheet plus ice shelf thinning. Would it be possible to separate it more from the other processes? The grounding line retreated significantly between

1996 and 2016 for this glacier and it would therefore be interesting to know how much the grounding line evolution contributed to the acceleration. The text about grounding line is also not always clear: it is not included as a separate process, but it is sometimes mentioned along other processes (e.g not mentioned in the abstract, previous studied about grounding line retreat only mentioned in the introduction, stated separately from the rest in conclusions, etc), so I would like to see addressed in a similar way throughout the manuscript. Part of this confusion might come from the problem that grounding line retreat is not clearly separated from other processes.

These experiments investigate the instantaneous response to changes in geometry. In some places of the text, it is clearly stated, but in other places it is not clearly stated. I would also like to see more discussion on this aspect to assess how this could impact the results, especially how the limitations in the model and observations could impact the results.

Minor comments

p.3 l.65: How linear do you expect the different processes to be? Is that something that can be investigated a bit more? Also, with what processes are the impact of grounding line changes included? Same for l.71-72.

p.3 l.74: Despite a lot of limitations, calving has been included in some models for a long time now (Martin et al., 2011) and a growing number of ice flow models are starting to include it at large scale in various forms (Seroussi et al., 2019).

p.3 l.75: I was wondering if this should not be: Delta Ua= Delta Uc=0

p.3 l.86: What knowledge is referred to here? As mentioned on l.87, it cannot be estimated from observations, I am wondering where this comes from?

p.5 Fig.1c: How do you explain the thinning just downstream of the 1996 grounding line?

p.6 l.151: U* is a little confusing, it might be more clear to list the perturbation experiments or explain a bit more what the asterisk refers to.

p.7 l.167: I would like to see more information on the inversion for A and C. What are the initial values, what are the minimum and maximum values admissible, . . . ?

p.7 l.173: What about the thermal regime? What is assumed for that? Many models of Pine Island have been developed over the past few years, so I guess it would not be difficult to use the ice thermal regime from a previous model to get a first guess of the rheology?

p.9 l.237: It is not clear which "earlier conclusion" is referred to here.

p.10 Figure 3e: How gate-dependent are the results? What would happen if gates a few kilometers upstream or downstream where used?

p.11 l.265: Is that what is expected in that case? Because the effect accumulates in time over 20 years and, even if it is reflected in the 2016 geometry, limitations in model parameters and errors in observations might limit the ability of the model to simulate good instantaneous response.

p.13 l.308: As mentioned above, you have here two sets of parameters (A2016 with C1996 and A1996 and C2016) that reproduce the 2016 velocity field. There is an infinite number of combinations that can reproduce a given field, so given the limited constraints provided for the inversion of A and C, can we make sure that the results are not dependent on the combination of parameters used?

p.17 l.397: The limitation of Weertman sliding is mentioned several times, but not really discussed, so it would be nice to see a bit more discussion on that.

p.17 l.403: It should be specified that this is an instantaneous response.

p.17 l.407: I thought that the results attributing the remaining changes to rigidity or friction were inconclusive as the fields computed were unphysical?

p.18 l.436: So what is used in this region and for the transition?

p.19 453: Why use a uniform value of -15 degrees?

p.19 l.463: What are the constraints used for A and C (minimum and maximum values possible)?

p.22 l.505: How about the relative weight for the two parts of the cost function referring to A and C? How are the weights calibrated?

Technical comments

p.4 l.121: we required -> we needed

p.4 l.122: How about grounding line positions?

p.6 l.160: It would be good to specify where you infer A because different groups use different parts of the domain (entire domain vs floating parts only)

p.8 l.198: missing Delta in front of UCalv or Delta UThinCalv)

p.11 l.262: it would be more informative to provide numbers about the different between Delta UThinCalv and Delta UCalv + Delta UThin rather than just saying they are about the same.

p.11 l.275 and l.277: hypotheses -> hypothesis

p.14 Figure 5d: Would it be possible to add results for more exponents? At least the value for m=21 to see the highest change.

p.17 l.406: increases -> increase

p.17 l.413: the DOI is missing for now.

p.18 l.338: the finite element method

p.18 l.438 to l.445: references are missing for the stress balance approximation, inverse capabilities, . . .

p.20 Figure B1c: It would be good to see the temperature equivalent to the rate factor

on top of the rate factor on the colorbar.

On all the spatial figures, the x and y axis are psx and psy, which looks a bit awkward to me, but that's not very important.

References

Martin et al., The Potsdam Parallel Ice Sheet Model (PISM-PIK) - Part 2: Dynamic equilibrium simulation of the Antarctic ice sheet, doi: 10.5194/tc-5-727-2011, The Cryosphere, 2011.

Seroussi et al., initMIP-Antarctica: an ice sheet model initialization experiment of IS-MIP6, doi:10.5194/tc-13-1441-2019, The Cryosphere, 2019.

---

## Author Comment (AC1) · 7 Oct 2020

We would like to extend a warm thank you to all the reviewers for their diligence in reviewing our manuscript and for their encouraging comments. We hope that referees find our responses and changes comprehensive and adequate. Original comments and our replies are presented in the attached supplement; they appear in black and blue italic text respectively. A revised manuscript with tracked changes is also included at the end.

Best regards,

Jan De Rydt, Ronja Reese, Fernando Paolo, Hilmar Gudmundsson

[Figure]

Please also note the supplement to this comment:
https://tc.copernicus.org/preprints/tc-2020-160/tc-2020-160-AC1-supplement.pdf

---

## Author Comment (AC2) · 7 Oct 2020

**Reply to reviewers - Drivers of Pine Island Glacier speed-up between 1996 and 2016.**

Jan De Rydt, Ronja Reese, Fernando S. Paolo, G. Hilmar Gudmundsson

October 7, 2020

**1 Reply to Anonymous Referee 1**

This is a nice paper showing some well-constructed model experiments designed to understand the sensitivity of PIG to various parameter. Despite my many comments and criticisms, I would very much like to see this paper published and I think it makes a valuable contribution to the field.

*We thank the reviewer for their kind support.*

I would like to see more discussion on the quality of the data and how it affects the results. To me the quality of the 1996 data from radar altimetry are highly suspect – ERS DEMS can be off by 10s or 100s of meters in places. Figure 1c does nothing to improve my confidence in these data, particularly the strong gradient in the thickness change across the ice shelf, with nearly zero thinning on the shelf centerline. Thus, I expect some model discrepancies could be explained by errors in the data sets used to constrain the model.

*As the reviewer correctly points out, satellite altimeters have limited accuracy, which means they are less reliable for DEMs. However, they are very precise instruments, which makes them excellent for change detection. That is why we did not use a snapshot DEM for the 1996 geometry (such as those based on ERS data), but compiled a time series of height changes from an extensive set of overlapping satellite altimeter data, as described in App.A. To derive net thickness changes between 1996 and 2016, we have integrated the altimeter trend over a 20-year time interval, which is very robust. To obtain ice thickness for 1996, altimeter thickness changes were subtracted from the Bedmachine reference thickness, based on REMA. The REMA DEM has very high resolution (good in the horizontal component), but has no way to constrain the vertical component of floating ice, as it is derived from optical sensors. So the surface elevation derived from REMA, like Bedmachine used here, is in fact tied to CryoSat-2. Appendix A in the main paper discusses a number of limitations of this approach, including uncertainties within a buffer along the grounding line. To gain additional confidence in our 1996 ice thickness distribution, we compared the grounding line location in the model to independent data from DInSAR (Rignot et al., 2014). We have added a new Figure to Appendix A (see Figure 1 below) to show that they agree well. The reviewer also highlights the 'nearly zero thinning on the ice shelf centerline'. The near-zero or positive thickness changes in this area are consistent with previously observed and published data, and we refer to e.g. Joughin et al. (2019) (their Figure 2 and supplementary data) and (Lhermitte et al., 2020) for more details.*

Related to this issue, I would rather see results like Figure 4b expressed as changes in basal shear stress. Particularly in light of the noisy 1996 data set, I would expect inferred changes in basal shear stress to be largely due to topographic (driving stress) errors.

*We hope that our discussion above has taken away the reviewer's concerns about the quality of the 1996 ice thickness data. With regards to Figure 4b, we present the corresponding changes in basal shear stress $\tau_b$ in Figure 2 below. The changes are largest in the vicinity of the grounding line, as expected, and there is no obvious correlation to noise in the topography. In particular, changes in driving stress along the central fast-flowing trunk are small, and most of the increased forward motion of the glacier ($U_b$ in Eq. 4 of the manuscript) is captured by changes in $C$, as shown in Figure 4b of the manuscript.*

As mentioned in some of the comments below, I have concerns about the joint inversion comment for A and C. While some prior info seems to be used, it's not well documented in the appendix. For

[Figure]

Figure 1: New figure added to Appendix A showing model and DInSAR grounding lines, altimeter data coverage for $\frac{dh}{dt}$ between 1996 and 2016 (dots and circles), and the total $\Delta H = H_{16} - H_{96} = \int_{96}^{16} \frac{dh}{dt} dt$ field used in this study (grey to blue background color).

example, do the p priors allow greater variation of A in the margins, while suppressing it elsewhere. I suspect not from the outputs. The inversions for A bear no resemblance to what my intuition would say the distribution should look like. There are numerous papers debating factors of 10 enhancement in shear margins, but this solution as irregular, patchy variations that vary by more than a factor of 100 (shouldn't the priors have not let A vary so much), sometimes with blobs of effectively very cold/stiff ice crossing the margins. I would really have liked the results better if the inversions for A had been restricted to the shelf where they don't make an already ill-posed problem even more so. This would have allowed for better discrimination of any role damage (A enhancement) on the ice shelf shear margins.

*We extended the accompanying text for the cost functions (Eq. B1 and B2) in the Appendix. In particular, we now stress that the regularization terms contain a gradient and an amplitude contribution, both of which are multiplied by a spatially constant prefactor. The role of the gradient premultiplier, $\gamma_s$, was discussed at length in Appendix D. However we did not elaborate on the dependency of our results on the amplitude premultiplier, henceforth called $\gamma_a$. The value of $\gamma_a$ was set to 1 throughout our simulations, but it is in fact a model parameter, which controls the weights of the C and A amplitude terms in the cost function. For example, $(g_a)_A \gg 1$ would force the final solution for A to be 'close' to the prior, hence suppressing any large spatial variations in A. To use a spatially varying $g_a$ to allow greater variations of A in some areas (such as the shear margins) based purely on heuristic arguments, seems rather ad-hoc. We therefore keep $g_a$ constant throughout the domain. An L-curve analysis, similar to Figure A1b for $\gamma_s$, was used to determine the optimal choice $\gamma_a = 1$. As the reviewer points out, some modeling groups do indeed set $A = A_{prior}$ for the grounded ice. In our view this does not make the ill-posed problem better behaved. Instead,*

[Figure]

Figure 2: Changes in basal traction for the $\mathcal{E}_C^3$ experiment.

*it assumes that one has perfect knowledge about A for the grounded ice (i.e. $A = A_{\text{prior}}$ exactly, everywhere). This is a strong assumption that might be far from the truth, as significant uncertainties in (depth-integrated) ice temperature, damage and fabric persist for the grounded ice, all of which should be captured by spatial variations in A. Given the range of physical properties that potentially influence A, it seems incorrect to interpret the latter in terms of temperature/damage alone, and the spatial distribution might therefore not always be intuitive. Nevertheless, the highest values of A in Figure A1 are primarily constrained to the shear margins, which is consistent with our expectations. To comprehensively address the ambiguities associated with different inversion methods and model parameters requires a full Bayesian treatment, which, we feel, is beyond the scope of this study.*

Other papers have presented similar results using a forward approach with somewhat better agreement with the data, though for a shorter time period (e.g., Gillet-Chaulet et al 2016 and Joughin et al 2019). There is some discussion about this work, but more comparisons could be drawn. For example, to get good agreement by assuming a reduction in basal traction near the grounding line driving by height-above flotation (effective pressure) variations. Could something like that be done here.

*We agree that Coulomb-limited sliding laws can have an important impact on the model response to changes in geometry, in particular in the vicinity of the grounding line, as demonstrated by e.g. Gillet-Chaulet et al. (2016) and Joughin et al. (2019). Although the analysis in this study has expanded the often-used Weertman law with $m = 3$ exponent to allow for a spatially heterogeneous exponent, we were unable to fully reproduce observed flow changes in the vicinity of the grounding line. Coulomb-limited sliding might lead to an improved fit, and although such experiments should be carried out as part of a separate study, we have expanded our discussion on this topic in Section 3.3.*

The colour maps are all in shades of red are hard to interpret. Maybe a different colour map or contours would help.

*We have tried to make our figures colorblind friendly and have experimented with various alternative colormaps, but did not find these to be particularly helpful. We have added a 50% contour to figures 3 and 5 to better guide the readers' eye.*

Line 27: Would be fair to cite Seroussi et al 2014 (doi:10.5194/tc-8-1699-2014) and Joughin et al, 2010 (in refs) in this list. Alternatively, these might be better in line 34.

*Thank you for suggesting these references; we have added both.*

Line 70-75. Where does loss of traction due grounding line fit it, which is caused by thinning (this is probably a bigger factor than the original loss of buttressing that triggered it – e.g., Figure 4 Joughin 2019 ref, which thinning alone can actually slow ice shelf a GL velocities without the loss of traction from ungrounding). Payne et al 2004 also look at the shift of the GL, rather than just the loss of ice shelf thinning.

*The loss of basal traction due to ungrounding is represented in experiments $\mathcal{E}_{\text{Thin}}^{m}$, which prescribes observed thickness changes of both the floating and grounded parts including any associated grounding line movement. We have clarified this point in the manuscript.*

Line 84 deltaUA+deltaUC=0. Again is the loss of traction as the grounding line retreats being bookkept as a change in deltaUC or a change deltaUthin. This is also important for interpreting line 95 because the paper cited shows the thinning driven response, but that includes loss of traction as the ice approaches flotation and finally ungrounds. All I am asking here for is a sentence clearly stating where the loss of traction due to ungrounding is bookkept.

*Thank you for asking us to clarify this important point. The loss of traction due to grounding line (GL) retreat is simulated in $\mathcal{E}_{\text{Thin}}^{m}$, which accounts for all observed thickness changes between 1996 and 2016 and therefore includes GL movement. We have added a sentence to emphasise this point. The modelled changes in GL position correspond almost exactly to observed changes from DInSAR, yet, changes in geometry (incl. loss of traction) were not generally sufficient to explain the speed-up of PIG between 1996 and 2016. At least not in our model setup with Weertman sliding exponent $m = 3$. To account for any remaining discrepancies between modeled and observed surface velocity in 2016, two possibilities were explored: experiment $\mathcal{E}_{C}^{3}$ was set up to determine what additional changes in the slipperiness would be required to account for the remaining changes in speed, whereas $\mathcal{E}_{A}^{3}$ was set up to determine changes in the rate factor required to account for the remaining changes in speed.*

Line 139. "The resulting values for $\Delta H$, linearly interpolated across the grounding line and in data sparse areas, are shown in Fig. 1c" I don't see how this is valid or maybe I am not understanding what is being done. You could have a case where the shelf thins by 10s or 100s of meters due to melting, but the grounded change in thickness is far less. Am I misinterpreting something.

*The data coverage used to derive thickness changes between 1996 and 2016 is shown in the new Figure A1 (see Fig.1 above). Within a 3 km buffer downstream of the grounding line we did not have any reliable altimetry measurements, and linear interpolation of $\Delta H$ accross this buffer could indeed lead to an underestimate of ice shelf thinning. For the remainder of the domain, data sparse areas largely correspond to recently ungrounded parts of the central glacier. If one assumes near-constant thinning of the ice shelf and a steady retreat of the grounding line in this area, linear interpolation of thickness changes seem appropriate. As an independent check, the resulting changes in grounding line position closely follow the DInSAR measurements.*

Line 165. "linear, viscous or close-to plastic" It should be "linear viscous, non-linear viscous, or close to plastic"

*Thank you for spotting this incorrect use of terminology.*

Line 167: "which caused small variations in $\tau_b$ between cases," This has to do with the interplay between A and taub, not so much the exponent. If you had used a fixed A, then taub is in fact what you are using to achieve stress balance with its parameterization via the sliding law (any sliding law). So, I am not sure the following sentence is correct and probably should be removed. What is different about the two studies is that A is determined on the entire domain for this study vs only on the shelf. A such additional degrees of freedom are introduced, which will give a better fit to the data, but could result in model parameters that are far from the true physicl values (eg. To stiff A balanced by a too slippery bed). Given that the response is dependent on the flow and sliding law parameters, the accuracy of the fit is somewhat immaterial. There is a counter argument that many other studies fit for one parameter or another using an assumed value for the other, so there could be errors in the assumed value. But I am not seeing a strong case for the improvement here.

*We have removed the sentence "We consider our approach to be more appropriate for this study, as our focus is primarily on an accurate model representation of the surface flow (e.g. Eq. 2).",*

*which also confused reviewer 2. We agree that both methods (i.e. separate inversions for different values of m vs. a single inversion for m = 1) are both valid and differences are irrelevant for our results.*

Line 172: I may be playing with semantics here but what is obtain is not a "best estimate" it's an estimate that produces the best fit, but as mentioned above, this by no means ensures that it is the best estimate.

*This is a very valid point. In fact, using 'an estimate that produces the best fit' is also debatable, as the inversion is more than just a fitting exercise. To be precise our statement should be phrased in the context of the cost function, which contains prior information and error estimates. In the interest of clarity we have decided to remove this sentence.*

Line 176 "basal slipperiness were kept fixed" Please how the ungrounded region is treated – either here or above. While it may seem obvious, when various parameters are being held fixed, it's not always immediately clear that the traction is being zeroed when the ice goes afloat.

*We will add a sentence here to explain that areas where changes in ice thickness cause ungrounding of the ice, this automatically implies that $\tau_b = 0$ and the value of $C$ becomes irrelevant. For areas that remain grounded, $C$ is kept fixed.*

Line 451: Why not initialize with an A estimated from temperature model; solutions are often better if you start nearer the answer. Also given that there are blobs of A in close proximity each other that can vary by a couple of orders of magnitude (a difference between ice at 0 and -30 deg) without any real correspondence to the flow pattern (outside of some shear margins) it would be worth some comment here about how physically realistic the solution is at least in a qualitative sense. Even for shear margins, the lower trunk on the west side where you would expect some enhancement, the ice actually looks like it's much stiffer than the reference -15 deg.

*Besides the sensitivity experiments presented in Appendix D, additional experiments were performed to test the robustness of the inversion results, in particular the A field. We did not include these results in the manuscript, but are happy to provide more details here. The first set of experiments, as outlined in one of our replies above, aims to test the sensitivity of A to the value of $(g_a)_A$ in the cost function. The pre-multiplier $(g_a)_A$ effectively controls how 'far' the solution A is allowed to deviate from its prior value $A_{prior}$. Results are summarized in Fig.3 (first row) and can be emphasised as follows: For $(g_a)_A \leq 1$, the solution for A and corresponding misfit (I, Eq.B1 in the main text) do not vary much. For $(g_a)_A \gg 1$, A is forced to be close to the prior value, as expected, and the misfit I increases. In the latter case, variability in the flow does not automatically get absorbed into larger variability of the slipperiness, C, and reducing $(g_a)_C$ does not change this conclusion. An L-curve approach was used to determine the optimal values $(g_a)_A = 1$ and $(g_a)_C = 1$ for our experiments. We are not aware of reliable depth-average ice temperature estimates that can be used as prior values in this region, and even if they existed, spatial variations of other ice properties (damage, fabric,...) upstream of the grounding line would still need to be inverted for. To partly address this issue, we assessed the sensitivity of A to the prior value in a the second set of experiments. Separate inversions were performed for $A_{prior} = -5C$, $-15C$ (the default) and $-25C$. Results (bottom row if Fig. 3) show that $A_{prior}$ does not have a significant effect (both qualitative and quantitative) on the final A field for ice that is not close-to-stagnant, and the spatial distribution of A is robust. Finally, we would like to mention that we have been pursuing some of these points in recent work on Thwaites Glacier. We kindly refer the reviewer to Barnes et al. (2020), currently in discussion, for a detailed analysis of inversion products for different models (ISSM, Úa and STREAMICE). some of these indeed use temperature estimates as part of the prior and/or assume perfect knowledge about A for grounded ice.*

Line 216: Would be nice to have a brief introduction to this section between 3 and 3.1. Actually this would be a good place to describe the flux gates, which break up the flow of the text below.

Line 232-237.5: Please consider moving this flux gate description to intro paragraph before 3.1 as described above.

*We thank the reviewer for this suggestion, and have moved our description of the flux gates to the start of Section 3.*

[Figure]

Figure 3: Sensitivity of *A* to variations in key inversion parameters.

Line 221: "viscous, rate-strengthening bed rheology " Please add "non-linear" viscous here. Also please note that this description applies under the assumption of till at the ice bed interface. As conceived, the original Weertman law applies to sliding over a hard, non-deforming bed, with ice and rock separated by a thin water film. Inversions here and elsewhere suggest both types of bed may be present beneath PIG. To the extent that its applicable to till, its likely only at low sliding speeds (see Zoet reference).

*We acknowledge the reviewer's earlier comment to use "linear viscous, non-linear viscous, or close to plastic" for the different types of rheology, and for stressing that a non-linear powerlaw describes both hard-bedded sliding (as originally conceived by Weertman), and till deformation, as outlined by Zoet and Iverson (2020).*

Line 224-225: 50% speedup. Be clear on where, it looks like this magnitude really applies only to the outer shelf. Might also be good to note that results are consistent with the Schmeltz et al 2002 reference that is cited in the text, where a similar experiment was performed (they only get a speedup of 40% right at the shelf front – mid shelf its more like 20%). Would be far more accurate to say "restricted to the OUTER ice shelf". A lack of speed from calving at the GL is consistent with a number of published results showing speed on PIG over the last decade, during which time there have been large calving events.

*We will add "OUTER shelf" to better pinpoint the geographical extent of instantaneous speed-up induced by calving, and refer to Schmeltz et al. (2002) to emphasize consistency with earlier studies.*

Line 237: "This supports..." This what? How about something like "This moderate response supports..." Now please cite the earlier works that your work supports.

*We have moved the preceding sentences about the flux gates to the start of the section, and have blended the last two sentences of this paragraph to say "Figure 3e shows that the retreat of the PIG calving front between 1996 and 2016 accounts for 2% and 13% of the observed flux changes through*

*Gate 1 and 2 respectively, which indicates minor instantaneous changes to the flow upstream of the grounding line."*

Lines 239-243: This text could be wordsmithed – there are least 3 place saying something about the similarity to calving response. Could just say something like "The responses to calving and thinning are similar at the flux gates, but the calving induces a larger response on the outer shelf (50% vs ¡ 25%). "Ice shelf thinning is generally accepted to be the main driver of ongoing mass loss of PIG, and patterns of ice shelf thinning elsewhere in Antarctica are strongly correlated to observed changes in grounding line flux (Reese et al., 2018; Gudmundsson et al., 2019)." This is a gross over-simplification of the conventional wisdom, which is that loss of buttressing can induce near GL thinning, which leads to retreat, which induces more loss. As AR5 notes, "Problems arise at the GL because, in addition to flotation, basal traction is dramatically reduced as the ice loses contact with the underlying bedrock (Pattyn et al., 2006). This is a topic of active research, and a combination of more complete modelling of the GL stress regime (Favier et al., 2012) and the use of high-resolution (subkilometre) models (Durand et al., 2009; Cornford et al., 2013) shows promise towards resolving these problems." Numerous studies have shown that the near GL speedup is a response to the loss of traction as the GL retreats, not the thinning itself, although that thinning believe to have triggered the whole process, likely through smaller speedups that thin the GL to flotation.

*We of course agree with the reviewer, and we are aware of the GL processes that cause glacier speed-up (this is the subject of our study). We do not believe that our statements are a "gross over-simplification". In particular, our sentence "the force perturbations that result from ice-shelf thinning alone, in particular the instantaneous reduction in back forces $\tau_{IS}$, are not sufficient to explain the magnitude of observed changes in upstream flow" echoes the reviewer's comments. We go on to explain that "time-evolving changes in geometry and mass redistribution upstream of the grounding line play a significant role in increasing the dynamic response of the glacier". In a revised version of the manuscript, we will add that the dynamic response includes grounding line movement and associated changes in basal traction.*

Line 246: "in upstream flow, CONSISTENT WITH OTHER STUDIES THAT HAVE SHOWN SIMILAR RESULTS [e.g., Payne 2004, Joughin 2010, 2019, Seroussi, 2014, Schoof 2007....].
*Done.*

Line 271 please add also "non-linear" before viscous. Also "non linear viscous bed rheology described by a Weertman sliding law". Weertman is not a viscous flow model even if the expressions are mathematically equivalent and generally is taken to mean an exponent between 2 and 3. Since your law applies to a broader range of exponents and indeterminate bed conditions (probably both till and hard bed), how about just substituting here and above power-law sliding in place of Weertman sliding law. At some point where you used m=3 you can say "power-law sliding with m=3, which for hard beds corresponds to a Weertman sliding law" to credit Weertman (a reference to his work would be nice).

*Thank you for clarifying these nuances in terminology. So far we have consistently used 'Weertman law' to refer to any sliding law that takes the form in Eq.4, irrespective of the exponent value. We have no objections against using 'Weertman law' for $m = 3$ only, and 'powerlaw' for $m \neq 3$. We will adjust the text accordingly.*

Line 290 "Weertman sliding" change to "power law sliding".
*Done.*

Line 300: Change "Weakening of the ice in these areas accounts for" to "Weakening of the ice in these areas is sufficient to account for"... Its only one model on the knob, so it's a sufficient but not necessary condition.
*Good point, thank you. We will changes this sentence as suggested.*

Line 302-303: I agree this change is physically improbable. But as noted above, so is the original A, which effectively has very warm ice in the margin just upstream of the margin where the ice very cold. See other comments about this.

*Thank you. We hope to have addressed the other comments in our earlier replies.*

Line 368: The result is also consistent with the Schoof type sliding law used by Joughin et 2019, which produces Coulomb plastic like behavior at speeds ¿ 300 m/yr and low-m (Weertman-like) behavior at slower speeds. The areas with plastic like behavior in Figure 6 transition to lower exponents in the area between the 600 and 100 m/yr contours shown in Fig 4, which would be worth commenting on.

*Thank you for pointing this out. We will draw further attention to the analogies between our work and (Joughin et al., 2009).*

Line 387: "Compared to spatially uniform values of m....". But this in effect what a sliding like that proposed by Schoof (and in another paper by Gagliardini et al) does – provides high-m behavior for fast flow and low-m behavior for slow flow.

*We agree. Our analysis has provided a way to bracket plausible values and a spatial distribution of m.*

Line 399: Change "Based on the most comprehensive observations..." By what metric? Other data sets are arguably more comprehensive (you have only 2 snapshots in time). Whether they are or are not the most comprehensive has no bearing on the value of this paper. Instead, simply say "Based on a comprehensive..." and you are on firm ground.

*We again apologize for the misunderstanding about our 1996 geometry. This is indeed a 'snapshot' geometry, i.e. with a fixed timestamp, but not based on 'snapshot' (ERS1-2) measurements. Instead, the 1996 DEM is based on a comprehensive timeseries of altimeter thickness changes, subtracted from the 2016 REMA DEM, referenced wrt Cryosat-2 data. Further details are provided in one of our earlier replies and Appendix A.*

Line 402: Remove "unprecedented". The are some nice results in this paper, but they largely echo the results of earlier work (I mean no slight here this is true of most papers).

*Done.*

Line 408: See general comments, but Joughin et al for a different time period got quite close agreement in a similar set of experiments by changes in basal shear stress proportional to the height above flotation in the region immediately above the grounding line. A compare and contrast sentence on this point would be good.

*We agree, and will add a sentence at the end of Section 3 to re-iterate this point.*

L410: we found that the recent increase in flow speed of Pine Island Glacier is only compatible with observed patterns of thinning if a heterogeneous, predominantly plastic bed underlies large parts of the central glacier and its upstream tributaries, CONSISTENT WITH EARLIER FINDING (there are several that are appropriate).

*We will add the suggested text here. References to stress consistency with earlier findings have been added at appropriate places in the manuscript, partly in response to the reviewer's comments, and we don't see any point in repeating this long list of references in the conclusions.*

**2  Reply to referee 2 Stephen Cornford**

This paper attributes the speed up in Pine Island Glacier over a 20 year period to a combination of ice thinning, calving front change and rheological change. It concludes (as have others) that the PIG trunk appears to be subject to Coulomb sliding, at least in some parts. This is a well written paper based on sound methods. I do have some minor issues with the manuscript.

*We thank the reviewer for their kind support.*

L65 (eqn 2). The right hand side does not make sense to me for the reasons you point out later: you can't determine these things individually and then add them up. You can certainly (as you do) look at each in turn, and even talk about combinations. I can see why you want a notation like this, but at the end of the day, the equation is not correct given the usual definition of addition.

*We originally added this equation to help us introduce the notations $\Delta U_{\mathrm{Calv}},...,$ but in hindsight we agree that this is incorrect and confusing. We have deferred the definitions of $\Delta U_{\mathrm{Calv}}$, etc. to Section 2, where it becomes clear within the model context what these changes in velocity are.*

L165 : The inverse problem is solved for each m, and Tb is slightly different. But that would not be so if the velocity were the same in each case, and the velocity could be the same in each case (in the optimal case, equal to the observations). The differences must be due to optimization method or similar. Joughin's method is equally valid, so you should not claim that yours is more appropriate.

*A similar point was raised by reviewer#1. We fully agree and have removed this statement from the ms.*

L30: "model studies have primarily focused on the important problem of simulating the response of PIG to a potential anthropogenic intensification of melt." Perhaps – but Joughin 2010 and 2019 were also concerned with attributing cause to observed change. This is mentioned later (from L42), so you have not neglected these papers, but it does make it makes 'primarily' incorrect

*Good point, and we will change the sentence: "...several model studies have focused on..."*

L95: Joughin et al. (2019) and Joughin 2010

*Reference added.*

2.2.1 Inverse experiments, and general use of inverse in a 'slang' fashion. These are optimization experiments, which involve solution of inverse problems. Ua does not have inverse capability, it has optimization capability.

*Agreed. We have changed the title of the subsection and replaced 'inverse' with 'optimization' elsewhere for consistency.*

Figure 3. Either have 'unaccounted' bars in every row, or none.

*Bars have been removed.*

L362. moptimal = f2/(f1-1) ? 100% ≠ 100.

*Indeed, we want $\Delta U_{\mathrm{CalvThin}}/\Delta U = 1$ and therefore $m_{\mathrm{optimal}} = f_2/(f_1 - 1)$. Thank you.*

L500 "In a Bayesian framework, the regularization plays the role of a prior and is added to the misfit, which corresponds to the likelihood". Tikhonov regularization can be seen as derived from Bayes rule with certain assumptions about the priors. But the methods used here are, I would say, not derived from Bayes rule because the prior is not determined from additional observation or theory: it is chosen to regularize the solution, and the relation to Bayes rule is incidental. Maybe a personal bugbear of mine.

*Thanks for pointing out this potential confusion. Since we do not employ the Bayesian framework, we have removed the comment.*

**3 Reply to Anonymous Referee 3**

The manuscript "Drivers of Pine Island Glacier retreat from 1996 to 2016" by J. De Rydt and colleagues analyses the role of several processes in causing the observed changes of Pine Island Glacier between 1996 and 2016 using numerical modeling. They estimate the relative role of calving, ice shelf thinning, ice shelf and ice sheet thinning, and well as combinations of these changes and their ability to reproduce the observed changes in ice velocity. The manuscript is well written, well explained, the figures are appropriate and this work is important as we try to better understand the glacier's changes in this region.

*We thank the reviewer for their kind support.*

There is one major point, however, that I would like to see addressed to make sure the results are robust and the conclusions not impacted by model parameter. The 1996 velocities are reproduced by inverting simultaneously the rigidity parameter A and the basal friction parameter C. This has been done in several studies, however there is an infinite combination of these two parameters that can yield similar velocity fields with drastically different values for each of the parameters. This

is something that is observed for the inversion of 2016 in the present manuscript: two additional inversions are done to fit the velocity observations for that year, one changing only C, and the other one changing only A. Therefore, two different combinations of parameters (A1996 with C2016 and A2016 with C1996) yield similar velocity fields in agreement with the 2016 observations. This is a recurrent problem in our field and the alternatives are limited, however I am wondering if a different combination of parameters would lead to different conclusions and I would like to see this point addressed. One solution to do so could be for the 1996 observations to first optimize the friction parameter and then the rigidity parameter in a first case, while a second case would first optimize the rigidity parameter and the friction afterwards. I expect these two cases to exhibit significant differences as most of the misfit will be captured by the first model parameter inferred, with the second one only capturing "residual-like" misfit. Experiments done with these two initial conditions will allow to make sure that the conclusions are robust and not impacted by the choice of these parameters. As explained in detail above, I would like to make sure that the conclusions are robust under a range of values for A and C as both parameters are unconstrained. Additional experiments with other values such as proposed above would help ensure that this is the case.

*We agree that the problem is ill-posed, and the solution is -to some degree- dependent on the optimization scheme, which is different for all models. Inverting first for A and then for C, or vice versa, as the reviewer suggests, has potential pitfalls. As the reviewer mentions, it pushes most of the explanation of observed velocities into one of the variables with the other variable only explaining misfits produced by this procedure in the first step. There is no physical basis to think that one variable A or C is of higher order importance in explaining observed velocities. Instead, we have experimented with different regularization parameters, as detailed in further replies below, to make sure that our results are robust. In particular, we have changed the relative weights of amplitude and gradient terms in the cost function, to force closer/looser agreement with prior values, and/or allow greater/smaller spatial variability. In the end, optimal values for the weights were obtained using an L-curve approach, as detailed in a later reply and App.B in the manuscript.*

The role of grounding line retreat is confusing. It is not part of ice shelf thinning, but it is included in the ice sheet plus ice shelf thinning. Would it be possible to separate it more from the other processes? The grounding line retreated significantly between 1996 and 2016 for this glacier and it would therefore be interesting to know how much the grounding line evolution contributed to the acceleration. The text about grounding line is also not always clear: it is not included as a separate process, but it is sometimes mentioned along other processes (e.g not mentioned in the abstract, previous studied about grounding line retreat only mentioned in the introduction, stated separately from the rest in conclusions, etc), so I would like to see addressed in a similar way throughout the manuscript. Part of this confusion might come from the problem that grounding line retreat is not clearly separated from other processes.

*Many thanks for raising this point of confusion. We changed the title to "Drivers of Pine Island Glacier **speed-up** between 1996 and 2016" to be clear about the actual aim of the study. Specifically, we prescribe observed changes in the geometry (ice thickness changes and calving) and diagnose the resulting velocity changes in the model. Grounding line retreat and associated loss of basal traction is therefore part of the $\mathcal{E}_{\text{Thin}}$ experiments. The effect of grounding line movement cannot be disentangled from changes in grounded ice thickness, as the grounding line moves due to changes in ice thickness at and upstream of the grounding line over time. However, the effect of reduction in basal traction in newly ungrounded regions could be disentangled from the effect of grounded thickness changes. This could be assesses by targeted new experiments, as the reviewer suggests, looking into the effect of basal traction reduction (i.e. zeroing basal traction where ice ungrounds) without changing the geometry. Such an experiment, albeit well-defined, will not change the main conclusions of our work, namely that the model is unable to reproduce observed changes in speed in response to ice thinning, GL retreat and calving (at least for $m = 3$). We already present a comprehensive set of experiments, which include the loss of basal traction as part of $\mathcal{E}_{\text{Thin}}$, and therefore prefer to keep the suggested experiments as part of a future study.*

These experiments investigate the instantaneous response to changes in geometry. In some places of the text, it is clearly stated, but in other places it is not clearly stated.

*Our approach is indeed different from a full transient simulation, whereby the model simulates both changes in ice geometry and velocity. Our experiments could be seen as a 'hybrid', whereby changes*

*in ice geometry are prescribed, but the model diagnoses changes in flow. As we can directly compare modelled and observed changes in flow, our experiments provide a well-constrained framework to validate the model response function, defined by the model physics (SSA, powerlaw sliding, Glen's rheology etc). We note that diagnostic experiments are justified by the instantaneity of the Stokes flow problem, i.e. the flow for a given geometry can be solved, given boundary conditions, without requirements of knowing the flow at any other time. The instantaneous response of velocities to the observed changes in the geometry is hence the response expected from a full dynamic simulation with geometry changes being consistent with observations.*

I would also like to see more discussion on this aspect to assess how this could impact the results, especially how the limitations in the model and observations could impact the results.

*We hope to address the reviewer's comment about the dependency of our results on uncertainties in the datasets and model parameters in further replies below. In summary, the general limitations of our datasets are now discussed in more detail in AppA (also see Fig.1 above) and the discussion about limitations of our optimization approach has been expanded (see futher replies below and App.B in the manuscript).*

p.3 l.65: How linear do you expect the different processes to be? Is that something that can be investigated a bit more? Also, with what processes are the impact of grounding line changes included? Same for l.71-72.

*We have removed this equation as it is misleading. The terms on the right hand side are not independent quantities, and $\Delta U$ cannot be expressed as a sum of dependent quantities. We have removed all references to Eq.(2) from the ms.*

p.3 l.74: Despite a lot of limitations, calving has been included in some models for a long time now (Martin et al., 2011) and a growing number of ice flow models are starting to include it at large scale in various forms (Seroussi et al., 2019).

*We certainly recognise the use of various calving parameterizations in models, and it is a growing field of research, which is encouraging to see. We have changed the sentence to "…whereas only a limited but growing number of ice flow simulations include parameterizations of calving."*

p.3 l.75: I was wondering if this should not be: Delta Ua= Delta Uc=0

*We have removed all instances of $\Delta U$, $\Delta U_{\text{Calv}}$,… from the introduction, and introduced this notation together with the model experiments in Sect.2, which we believe is more appropriate.*

p.3 l.86: What knowledge is referred to here? As mentioned on l.87, it cannot be estimated from observations, I am wondering where this comes from? p.5 Fig.1c: How do you explain the thinning just downstream of the 1996 grounding line?

*We have reformulated this sentence, in line with our previous reply. "Knowledge" has been replaced by "Model estimates of".*

p.6 l.151: U* is a little confusing, it might be more clear to list the perturbation experiments or explain a bit more what the asterisk refers to.

*We have removed $U_*$ and referred to individual velocities ($U_{Calv}$,…) instead.*

p.7 l.167: I would like to see more information on the inversion for A and C. What are the initial values, what are the minimum and maximum values admissible, … ?

*We have added further details about the optimization to App.B*

p.7 l.173: What about the thermal regime? What is assumed for that? Many models of Pine Island have been developed over the past few years, so I guess it would not be difficult to use the ice thermal regime from a previous model to get a first guess of the rheology?

*In our study we also invert for A over the grounded parts of PIG to include - together with ice temperatures - further factors influencing the ice rheology such as impurities, damage,…(see e.g. the recent study by Lhermitte et al. (2020) who show extensive damage upstream of the GL). We agree that it would be interesting to compare the results with results from a thermodynamic study but since the interpretation of A cannot be based on ice temperature alone, this is also not straightforward. We leave this for a future study.*

p.9 l.237: It is not clear which "earlier conclusion" is referred to here.

*We have reworded this sentence and restructured this section. The new sentence reads "Figure 3e shows that the retreat of the PIG calving front between 1996 and 2016 accounts for 2% and 13% of the observed flux changes through Gate 1 and 2 respectively, which indicates minor instantaneous changes to the flow upstream of the grounding line."*

p.10 Figure 3e: How gate-dependent are the results? What would happen if gates a few kilometers upstream or downstream where used?

*Results are robust with respect to small (∼kilometer) shift in the position of the gates. In fact, the gates do not provide any information that isn't contained in the 2d maps of relative change (e.g. Fig3a-d), but were rather introduced as a convenient way to synthesise the results. The 2d maps provide all the details, and the reviewer is hopefully convinced that results for $(U_{\text{pert}}-U_{96})/(U_{16}-U_{96})$ (as in Fig3a-d) are spatially coherent and small shifts in the position of the gates will not greatly affect the values in Fig3e.*

p.11 l.265: Is that what is expected in that case? Because the effect accumulates in time over 20 years and, even if it is reflected in the 2016 geometry, limitations in model parameters and errors in observations might limit the ability of the model to simulate good instantaneous response.

*We are unclear what 'limitations' the reviewer refers to. Yes, there might be errors in the 1996 and 2016 ice thickness datasets that influence our results, but it is hard to see how this could account for an additional 72% (Gate1) and 36% (Gate2) of speed-up. Given the excellent agreement between model and DInSAR grounding line locations in both years (see Fig.1), we are confident that both thickness datasets are reasonably accurate. We also note that both thickness datasets have the same variability at small (∼km) spatial scales: the 2016 dataset is based on REMA, whereas the 1996 dataset was obtained by subtracting spatially smooth thickness changes (Fig.1). Small-scale errors in the thickness distributions are therefore "consistent" between 1996 and 2016 (i.e. errors related to the REMA dem), and modeled changes in ice velocities are entirely dictated by the spatially smooth thickness changes (i.e. errors in the small-scale thickness distribution cancelling out). We agree that in addition to the geometry, physical properties of the ice (rate factor) and bed (slipperiness) could change between 1996 and 2016, but we believe these have been addressed at length in the manuscript.*

p.13 l.308: As mentioned above, you have here two sets of parameters (A2016 with C1996 and A1996 and C2016) that reproduce the 2016 velocity field. There is an infinite number of combinations that can reproduce a given field, so given the limited constraints provided for the inversion of A and C, can we make sure that the results are not dependent on the combination of parameters used?

*We agree that the inverse problem is generally ill-posed. To partly address the reviewer's concern, we have carried out a range of sensitivity tests for model parameters that define the cost function (see our replies to reviewer#1, and replies to your other comments). With regards to the solutions $\tilde{A}_{16}$ and $\tilde{C}_{16}$ of the $\mathcal{E}_A$ and $\mathcal{E}_C$ experiments, these are 'end members' that bracket the maximum amount of change required to fully describe the observed speed-up. In reality, it is likely that A and C will evolve simultaneously, and if we assume that our inversions do a perfect job, the original solution $(A_{96}, C_{96})$ will seamlessly evolve to $(A_{16}, C_{16})$. However, if we do not allow C to change over time and keep it fixed at $C_{96}$, then experiment $\mathcal{E}_A$ diagnoses the required changes to A (solution $\tilde{A}_{16}$). Similarly, if A is not allowed to change over time and kept fixed at $A_{96}$, experiment $\mathcal{E}_C$ diagnoses the required changes to C (solution $\tilde{C}_{16}$). We hope this clarifies the rationale behind these experiments.*

p.17 l.397: The limitation of Weertman sliding is mentioned several times, but not really discussed, so it would be nice to see a bit more discussion on that.

*In reply to this comment and comments from reviewer#1, we have expanded our discussion about the limitations of a power law sliding at various places in the ms.*

p.17 l.403: It should be specified that this is an instantaneous response.

*In our study, we let nature carry out the transient evolution of the geometry (i.e we use observations of ice thinning and calving), but use a model to diagnose the corresponding velocity response. This is indeed different from a full transient simulation, where the model simulates both changes*

*in ice geometry and velocity. Our experiments could be seen as a 'hybrid', whereby changes in ice geometry are prescribed, but the model calculates the changes in flow. We are reluctant to call this 'instantaneous', but we think "...a modern-day ice flow model to diagnose dynamic changes in response to prescribed geometric perturbations" is more accurate.*

p.17 l.407: I thought that the results attributing the remaining changes to rigidity or friction were inconclusive as the fields computed were unphysical?

*We cannot conclude with certainty that simulated changes in A are indeed unphysical since they could represent damage, changes in fabric etc that are difficult to constrain or observe. We therefore prefer to keep the statement that 'large' changes are needed, without drawing any strong conclusions about their physical validity.*

p.18 l.436: So what is used in this region and for the transition?

*A linear interpolation of $\Delta H$ was used in data sparse regions. Further details have now been included in the App.A, including a new figure (similar to Fig.1 above).*

p.19 453: Why use a uniform value of -15 degrees?

*This is of course a somewhat arbitrary choice, based on reasonable assumptions about the dept-averaged ice temperature throughout the domain. To test the sensitivity of our results to the choice of prior value, we have performed two sets of additional experiments: (i) inversions were repeated for $A_{prior} = -25C$ and $A_{prior} = -5C$; results are shown in Fig.3 above and discussed in an earlier reply to reviewer#1. (ii) inversion were repeated for different values of $(g_a)_A$, which is a pre-multiplier in the amplitude regularization term in the cost function. The value of $(g_a)_A$ effectively controls how 'far' the solution A can deviate from its prior value. Results are shown in Fig.3 and discussed further in our reply to your question below, as well as in an earlier reply to reviewer#1.*

p.19 l.463: What are the constraints used for A and C (minimum and maximum values possible)?

*For numerical reasons, solutions for A and C are restricted to the interval [1e-100 1e100], but the solution is well within these limits.*

p.22 l.505: How about the relative weight for the two parts of the cost function referring to A and C? How are the weights calibrated?

*We have not explored relative weights of the A and C pre-multipliers in great detail. In the main experiments, our choice $(g_s)_A = (g_s)_C = 25,000$ was based on the L-curve analysis presented in FigB1. Similarly, we used an L-curve analysis to set $(g_a)_A = (g_a)_C = 1$, which are the pre-multipliers in the amplitude term of the regularization. In fact, we did not elaborate on the $g_a$ values in the main text, but in response to this comment and a related comment by reviewer#2, we have included further details in App.B. We have also carried out experiments for different values of $(g_a)_A \in [0.1, 1, 100]$, whilst keeping $(g_a)_C = 1$ constant, to allow for larger $((g_a)_A = 0.1)$ or smaller $((g_a)_A = 100)$ deviation from $A_{prior}$. Results for these experiments are shown in Fig.3 above, and described in our replies to reviewer#1.*

p.4 l.121: we required - we needed
*Done*

p.4 l.122: How about grounding line positions?

*Changes in GL position are embedded in the differences in ice thickness between 1996 and 2016. However, as the latter were constrained by independent DInSAR GL datasets, we have mentioned GL positions here as well.*

p.6 l.160: It would be good to specify where you infer A because different groups use different parts of the domain (entire domain vs floating parts only)

*We have added further details about the optimization to App.B*

p.8 l.198: missing Delta in front of UCalv or Delta UThinCalv)
*Well spotted, thank you.*

p.11 l.262: it would be more informative to provide numbers about the different between Delta UThinCalv and Delta UCalv + Delta UThin rather than just saying they are about the same.

*In response to a comment by reviewer#2, we have removed any equations involving $\Delta U$, as these can be confusing. The percentages in Fig.3e provide a quantitative measure for how 'additive' the experiments are. E.g., flux changes through gates 1 and 2 in the ISThin and Thin experiments add up to 27% and 58% respectively, whereas flux changes in CalvThin are 28% and 64% respectively. The flux response in the combined experiment is therefore somewhat larger than the added response of each individual experiment, but we believe these differences are insignificant and not worth elaborating on.*

p.11 l.275 and l.277: hypotheses - hypothesis
*Done.*

p.14 Figure 5d: Would it be possible to add results for more exponents? At least the value for m=21 to see the highest change.

*We believe that additional panels for higher m are not particularly insightful, as changes relative changes to the velocity response become smaller for increasing m. In other words, a plot for $m = 21$ would look rather similar to the plot for $m = 13$. This can also be seen from Figure C1a, which shows the asymptotic behaviour in the response for higher values of m.*

p.17 l.406: increases - increase
*Done.*

p.17 l.413: the DOI is missing for now.
*A DOI will be provided with the final submission.*

p.18 l.338: the finite element method
*Done.*

p.18 l.438 to l.445: references are missing for the stress balance approximation, inverse capabilities, ...

*We have added references to (Hutter, 1983; MacAyeal, 1989) and (MacAyeal, 1992) respectively.*

p.20 Figure B1c: It would be good to see the temperature equivalent to the rate factor on top of the rate factor on the colorbar. On all the spatial figures, the x and y axis are psx and psy, which looks a bit awkward to me, but that's not very important.

*We have added temperature labels to the colorbar. We prefer to use psx and psy in all figures because this identifies the Cartesian coordinates system (polar stereographic).*

**References**

Barnes, J. M., dos Santos, T. D., Goldberg, D., Gudmundsson, G. H., Morlighem, M., and De Rydt, J.: The transferability of adjoint inversion products between different ice flow models, The Cryosphere Discussions, 2020, 1–32, 
[revised manuscript text omitted]

---

## Author Response (AR1)

**Reply to editor - Drivers of Pine Island Glacier speed-up between 1996 and 2016.**

Jan De Rydt, Ronja Reese, Fernando S. Paolo, G. Hilmar Gudmundsson

November 11, 2020

Dear Andreas,

Thank you for your response and the opportunity to submit a revised manuscript. We greatly appreciate your detailed report, and value your request to make additional changes to the manuscript. A point-by-point response to your comments and list of substantial reviewer comments with reference to changes in the manuscript are provided below. Our replies are in *blue italic* and all line numbers refer to the manuscript with tracked changes, included at the end of this document. We hope to have addressed all questions and concerns more explicitly now.

Yours sincerely,

Jan De Rydt on behalf of the authors

**Reply to editor comments**

Editor: "Dear Jan de Rydt and co-authors, Your manuscript received 3 thorough and in general positive reviews that highlighted the quality and originality and the relevance and added value of the manuscript with regard to analysis and understanding of the observed dynamic changes at Pine Island Glacier over the last two decades. A all referees agreed that this research and manuscript would valuable contribution to TC and should be published after some revisions.

However, besides quite a few rather minor editing corrections the referees also raised some more substantial points which mainly concerned in brief:
-critical points and further discussion on the joined optimization of A (ice rheology) and C (slipperiness)
-related the robustness of the inversions
-clarification of data quality
-clarification of interaction of grounding line and thinning in optimization
-aspects of eqn 2
-some updating and improvement in the reference to existing literature

*Below we explain how and where we have addressed each of these substantial points in the manuscript. For ease of reference we number them as follows:*

- *C1: further discussion on the optimization of A and C (raised by Refs 1 and 3).*

- *C2: robustness of the inversions (raised by Refs 1 and 3)*

- *C3: quality of the 1996 ice thickness data (raised by Ref 1)*

- *C4: clarification on how grounding line movement and associated loss of basal traction is included in our experiments (raised by Ref 1 and 3)*

- *C5: aspects of eqn 2 (raised by Ref 2)*

- *C6: improved references to existing literature (raised by Ref 1)*

*We have addressed C5 and C6 at length in previous replies to the reviewers, i.e. we decided to remove eqn 2 and have expanded references to existing literature throughout the manuscript. Our replies to C1-4 are detailed below.*

Editor: "Given the authors response to the reviews, it seems the minor and more substantial points are addressable by the authors or have already been addressed very well (see track change document) and the manuscript will likely get into a state to be accepted for publication.

Based on the response there are, however, a few points where the authors have not undertaken revisions or the authors disagree to the referees, and in the response they have given a convincing and detailed explanations or justifications to their decision, which is fine. However, in my view these answers (or parts thereof) may also be relevant for the general reader of the manuscript and not just the referees, and some explicit but very brief additional explanations/justifications (or links to the supplements) in the main text may be helpful. For example, a lengthy explanation has been given to the referees on the point of the joined optimization od A and C, but it is not clear/explicit if any of these explanations are planned to be incorporated into the manuscript. These explanations/justifications/clarifications should not just go to the referees, but some aspects of it may well be interesting for the general reader and should perhaps be considered in the main text. Specifically, for the case of the joined A and C optimization (Ref 1 and 3), some additional figures and clarifying text have been added in the supplement B2 which is helpful, but I think in the main text an additional reference to these add-ons in the supplements and maybe a brief discussion in the main text clarifying this point would be useful."

*In reply to C1 and C2: we have made a number of changes to motivate and better explain our choice of optimization scheme and optimization parameters. In particular, lines 205-208 (refers to the tracked changes at the end of this document) now include a word of caution about the solutions for A and C, and we explicitly refer to appendices B and D for a detailed discussion about the choice of optimization parameters and robustness of our results with respect to this choice. In turn, we have expanded Appendix B to highlight the role of the regularization multipliers (lines 535-546) including our motivation to invert for A and C across the full domain. We have also added some further context to our choice of prior information for A in lines 547-558.*

Editor: "To Ref1 comment 1: quality RS-data: a new figure has been added in the supplement, but in the main text, a reference to this figure and a short comment to it would be useful (relative dh change accurate)."

*In reply to C3: we have rewritten this part of the methods, see lines 145-171. We highlight our use of elevation changes (rather than a snapshot DEM) to estimate the ice thickness in 1996, and point out that this new approach is more accurate. We now refer the reader to the newly added figure A1 in the main manuscript, both in the context of the dh/dt data coverage (lines 161-165) and the good agreement between model and DInSAR grounding line locations in 1996 (line 167-169).*

Editor: "To Ref1 comment to line 139: data coverage: refer to this fig. A1 also in the main text."

*See previous reply.*

Editor: "clarification of interaction of grounding line and thinning in optimization"

*In reply to C4: Throughout the manuscript we have clarified that the 'thinning' experiment (described lines 229-231) includes grounding line movement and therefore implies the loss of traction in newly ungrounded areas. We refer to lines 76-77, 215-216, 239-241, 264-265 and 303 in the manuscript with tracked changes, as well as the caption of Fig. 3.*

Editor: "To Ref1 comment to line 451 and similar Ref3 (robustness of inversion results): the essence of this response would probably also be relevant for the general reader (not just referee) and could be briefly included in the main manuscript."

*In reply to C2: We prefer to keep details and discussions about the model configuration and optimization procedure in one place, and have therefore expanded Appendix B instead of the main manuscript. We have commented on the robustness of our results with respect to the choice of a priori values for A (lines 549-551) and have better motivated our choice of spatially constant regularization pre-multipliers (lines 541-546) and a priori values for A (lines 551-554). We comment that our results are robust across a range of pre-multiplier and a priori values, and refer to Appendix D*

*for a detailed discussion about the dependency on $\gamma_s$, which multiplies the gradient term in the regularization.*

Editor: "For the ease of reference to the author response by the editor it maybe useful to perhaps number the responses the major referee comments (e.g. R1.3....)."

*By numbering the main reviewer comments by C1-6 (see above) we hope to have addressed this suggestions by the editor.*

[revised manuscript text omitted]